# CHOREOGRAPHER: LEARNING AND ADAPTING SKILLS IN IMAGINATION

**Pietro Mazzaglia** [*] [1]    **Tim Verbelen** [1]    **Bart Dhoedt** [1]    **Alexandre Lacoste** [2]    **Sai Rajeswar** [2]

[1] Ghent University - imec    [2] ServiceNow Research

## ABSTRACT

Unsupervised skill learning aims to learn a rich repertoire of behaviors without external supervision, providing artificial agents with the ability to control and influence the environment. However, without appropriate knowledge and exploration, skills may provide control only over a restricted area of the environment, limiting their applicability. Furthermore, it is unclear how to leverage the learned skill behaviors for adapting to downstream tasks in a data-efficient manner. We present Choreographer, a model-based agent that exploits its world model to learn and adapt skills in imagination. Our method decouples the exploration and skill learning processes, being able to discover skills in the latent state space of the model. During adaptation, the agent uses a meta-controller to evaluate and adapt the learned skills efficiently by deploying them in parallel in imagination. Choreographer is able to learn skills both from offline data and by collecting data simultaneously with an exploration policy. The skills can be used to effectively adapt to downstream tasks, as we show in the URL benchmark, where we outperform previous approaches from both pixels and states inputs. The learned skills also explore the environment thoroughly, finding sparse rewards more frequently, as shown in goal-reaching tasks from the DMC Suite and Meta-World.

**Project website:** `https://skillchoreographer.github.io/`

## 1 INTRODUCTION

Deep Reinforcement Learning (RL) has yielded remarkable success in a wide variety of tasks ranging from game playing (Mnih et al., 2013; Silver et al., 2016) to complex robot control (Smith et al., 2022; OpenAI et al., 2019). However, most of these accomplishments are specific to mastering a single task relying on millions of interactions to learn the desired behavior. Solving a new task generally requires to start over, collecting task-specific data, and learning a new agent from scratch.

Instead, natural agents, such as humans, can quickly adapt to novel situations or tasks. Since their infancy, these agents are intrinsically motivated to try different movement patterns, continuously acquiring greater perceptual capabilities and sensorimotor experiences that are essential for the formation of future directed behaviors (Corbetta, 2021). For instance, a child who understands how object relations work, e.g. has autonomously learned to stack one block on top of another, can quickly master how to create structures comprising multiple objects (Marcinowski et al., 2019).

With the same goal, unsupervised RL (URL) methods aim to leverage intrinsic motivation signals, used to drive the agent's interaction with the environment, to acquire generalizable knowledge and behaviors. While some URL approaches focus on exploring the environment (Schmidhuber, 1991; Mutti et al., 2020; Bellemare et al., 2016), *competence-based* (Laskin et al., 2021) methods aim to learn a set of options or skills that provide the agent with the ability to control the environment (Gregor et al., 2016; Eysenbach et al., 2019), a.k.a. empowerment (Salge et al., 2014).

Learning a set of options can provide an optimal set of behaviors to quickly adapt and generalize to new tasks (Eysenbach et al., 2021). However, current methods still exhibit several limitations. Some of these are due to the nature of the skill discovery objective (Achiam et al., 2018), struggling to capture behaviors that are natural and meaningful for humans. Another major issue with current

---

[*]Correspondence to: pietro.mazzaglia@ugent.be.

methods is the limited exploration brought by competence-based methods, which tend to commit to behaviors that are easily discriminable but that guarantee limited control over a small area of the environment. This difficulty has been both analyzed theoretically (Campos et al., 2020) and demonstrated empirically (Laskin et al., 2021; Rajeswar et al., 2022).

A final important question with competence-based methods arises when adapting the skills learned without supervision to downstream tasks: how to exploit the skills in an efficient way, i.e. using the least number of environment interactions? While one could exhaustively test all the options in the environment, this can be expensive when learning a large number of skills (Eysenbach et al., 2019) or intractable, for continuous skill spaces (Kim et al., 2021; Liu & Abbeel, 2021a).

In this work, we propose **Choreographer**, an agent able to discover, learn and adapt unsupervised skills efficiently by leveraging a generative model of the environment dynamics, a.k.a. world model (Ha & Schmidhuber, 2018). Choreographer discovers and learns skills in imagination, thanks to the model, detaching the exploration and options discovery processes. During adaptation, Choreographer can predict the outcomes of the learned skills' actions, and so evaluate multiple skills in parallel in imagination, allowing to combine them efficiently for solving downstream tasks.

**Contributions.** Our contributions can be summarized as follow:

- We describe a general algorithm for discovering, learning, and adapting unsupervised skills that is exploration-agnostic and data-efficient. (Section 3).
- We propose a *code resampling* technique to prevent the issue of index collapse when learning high-dimensional codes with vector quantized autoencoders (Kaiser et al., 2018), which we employ in the skill discovery process (Section 3.2);
- We show that Choreographer can learn skills both from offline data or in parallel with exploration, and from both states and pixels inputs. The skills are adaptable for multiple tasks, as shown in the URL benchmark, where we outperform all baselines (Section 4.1);
- We show the skills learned by Choreographer are effective for exploration, discovering sparse rewards in the environment more likely than other methods (Section 4.2), and we further visualize and analyze them to provide additional insights (Section 4.3).

## 2 PRELIMINARIES AND BACKGROUND

**Reinforcement Learning (RL).** The RL setting can be formalized as a Markov Decision Process, where we denote observations from the environment with $x_t$, actions with $a_t$, rewards with $r_t$, and the discount factor with $\gamma$. The objective of an RL agent is to maximize the expected discounted sum of rewards over time for a given task, a.k.a. the return: $G_t = \sum_{k=t+1}^{T} \gamma^{(k-t-1)} r_k$. We focus on continuous-actions settings, where a common strategy is to learn a parameterized function that outputs the best action given a certain state, referred to as the actor $\pi_\theta(a|x)$, and a parameterized model that estimates the expected returns from a state for the actor, referred to as the critic $v_\psi(x)$. The models for actor-critic algorithms can be instantiated as deep neural networks to solve complex continuous control tasks (Haarnoja et al., 2018; Lillicrap et al., 2016; Schulman et al., 2017).

**Unsupervised RL (URL).** In our method, we distinguish three stages of the URL process (Figure 1). During stage *(i) data collection*, the agent can interact with the environment without rewards, driven by intrinsic motivation. During stage *(ii) pre-training*, the agent uses the data collected to

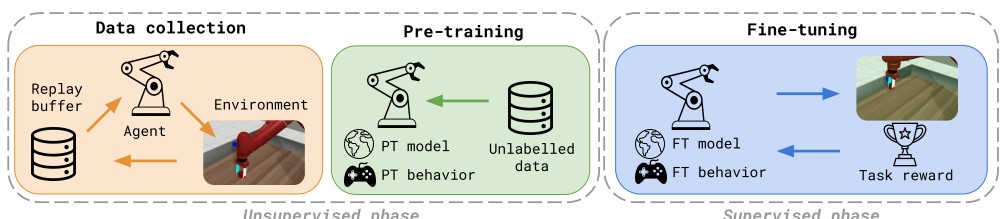

Figure 1: **Unsupervised Reinforcement Learning.** The agent should effectively leverage the unsupervised phase, consisting of the data collection and the pre-training (PT) stages, to efficiently adapt during the supervised phase, where the agent is fine-tuned (FT) for a downstream task.

learn useful modules (e.g. a world model, skills, etc.) to adapt faster to new scenarios. Crucially, stage *(i)* and *(ii)* need to happen in parallel for most of the previous competence-based methods (Eysenbach et al., 2019; Liu & Abbeel, 2021a; Gregor et al., 2016; Achiam et al., 2018), as skills discriminability is tied to the data collection process. Finally, in stage *(iii) fine-tuning*, the agent can interact with the environment receiving task reward feedback and should leverage the pre-trained modules to adapt faster to the given task.

**Mutual Information Skill Learning.** Option discovery can be formalized as learning a skill representation $z$ to condition skill policies $\pi(a|x, z)$. The objective of these policies is to maximize the mutual information between skills and trajectories of environment observations $\tau_x = \{x_t, x_{t+1}, ...\}$:

$$I(\tau_x; z) = H(\tau_x) - H(\tau_x|z) \qquad \textit{// forward} \tag{1}$$

$$= H(z) \ - H(z|\tau_x) \qquad \textit{// reverse} \tag{2}$$

The skill policy $\pi(a|x, z)$ results in a family of diversified policies with predictable behavior. Some previous work uses the forward form, either considering the entropy $H(\tau_x)$ fixed (Campos et al., 2020) or estimating it through sampling-based techniques (Sharma et al., 2020; Liu & Abbeel, 2021a; Laskin et al., 2022), and learning a function mapping skills to states to estimate conditional entropy. Other works use the reverse form, considering a fixed distribution over skills, with maximal entropy $H(z)$, and learning a skill discriminator that maps states to skills, to estimate conditional entropy (Achiam et al., 2018; Gregor et al., 2016; Eysenbach et al., 2019; Hansen et al., 2020).

## 3 CHOREOGRAPHER

Choreographer is a model-based RL agent designed for URL, leveraging a world model for discovering, learning, and adapting skills in imagination. In addition to a world model, the agent learns: a) a codebook of skill vectors, clustering the model state space, b) skill-conditioned policies, maximizing the mutual information between latent trajectories of model states and skill codes, and c) a meta-controller, to coordinate and adapt skills for downstream tasks, during fine-tuning. An overview of the approach is illustrated in Figure 2 and a detailed algorithm is presented in Appendix G.

### 3.1 WORLD MODEL

Choreographer learns a task-agnostic world model composed of the following components:

$$\text{Posterior:} \quad q_\phi(s_t|s_{t-1}, a_{t-1}, x_t), \qquad \text{Prior:} \quad p_\phi(s_t|s_{t-1}, a_{t-1}), \qquad \text{Reconstruction:} \quad p_\phi(x_t|s_t).$$

During the fine-tuning stage, the agent also learns a reward predictor $p_\phi(r_t|s_t)$. The model states $s_t$ have both a deterministic component, modeled using the recurrent state of a GRU (Cho et al., 2014), and a discrete stochastic component (Hafner et al., 2021). The remaining modules are multi-layer perceptrons (MLPs) or CNNs for pixel inputs (LeCun et al., 1999). The model is trained end-to-end by optimizing an evidence lower bound on the log-likelihood of data:

$$\mathcal{L}_{\text{wm}} = D_{\text{KL}}[q_\phi(s_t|s_{t-1}, a_{t-1}, x_t)\|p_\phi(s_t|s_{t-1}, a_{t-1})] - \mathbb{E}_{q_\phi(s_t)}[\log p_\phi(x_t|s_t)], \tag{3}$$

where sequences of observations $x_t$ and actions $a_t$ are sampled from a replay buffer.

### 3.2 SKILL DISCOVERY

To discover skills that are exploration-agnostic, i.e. do not require the agent to deploy them in order to evaluate their discriminability, we turn the skill discovery problem into a representation learning one. We learn a skill discrete representation $z$ on top of the model states using a VQ-VAE (van den Oord et al., 2017). This representation aims to encode model states into the $N$ codes of a codebook, learning $p(z|s)$, as well as the inverse relation, decoding codes to model states, via $p(s|z)$.

In the skill auto-encoder, model states are first processed into embeddings using a skill encoder $E(s)$. Then, each embedding is assigned to the closest code in the codebook $z_q^{(i)}, i \in 1...N$, through:

$$\text{quantize}(E(s)) = z_q^{(k)} \qquad k = \arg\min_j \|E(s) - z_q^{(j)}\|_2 \tag{4}$$

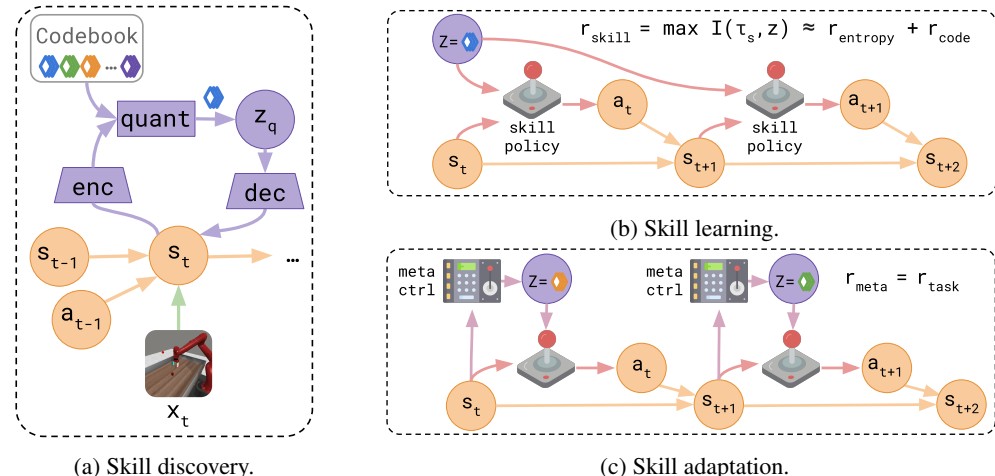

(a) Skill discovery.

(b) Skill learning.

(c) Skill adaptation.

Figure 2: **Choreographer.** (a) The agent leverages representation learning to learn a codebook of skill vectors, summarizing the model state space. (b) Skill policies are learned in imagination, maximizing the mutual information between latent trajectories of model states and skill codes. (c) The meta-controller selects the skills to apply for downstream tasks, evaluating and adapting skill policies in imagination by hallucinating the trajectories of states and rewards they would obtain.

The indices are mapped back to their corresponding vectors in the codebook, from which the decoder $D(z_q^k)$ reconstructs the corresponding model states. The training objective is:

$$\mathcal{L}_z = \|s - D(z_q^{(k)})\|_2^2 + \beta\|\text{sg}(z_q^{(k)}) - E(s)\|_2^2, \tag{5}$$

where $\text{sg}(\cdot)$ indicates to stop gradients. The first loss term is the reconstruction error of the decoder, while the second term encourages the embedded vectors to stay close to the codes in the codebook.

Crucially, as opposed to most previous work (van den Oord et al., 2017; Razavi et al., 2019), we choose to learn a single-code representation, where each model state gets assigned only one vector in the codebook. Our skill codes can thus be seen as centroids partitioning the model state space into $N$ clusters (Roy et al., 2018), so that intuitively each skill occupies one area of the state space.

**Code resampling.** When training single-code representations with VQ-VAE, the codebook may suffer from *index collapse* (Kaiser et al., 2018), where only a few of the code vectors get trained due to a preferential attachment process. One way to mitigate this issue is to use sliced latent vectors, which assign multiple smaller codes to the same embedding (Kaiser et al., 2018; Razavi et al., 2019), but this would imply losing the one-to-one association between codes and skills.

In order to mitigate the issue we propose a *code resampling* technique which consists of two steps:

1. keeping track of the inactive codes during training, which is achieved by looking at the codes to which no embedding is assigned for $M$ consecutive training batches;
2. re-initializing the inactive codes, with values of embeddings $E(s)$ from the latest training batch, selected with a probability $\frac{d_q^2(E(s))}{\sum_s d_q^2(E(s))}$, where $d_q(\cdot)$ is the euclidean distance from the closest code in the codebook, i.e. $d_q(E(s)) = \min_{i \in 1...N} \|E(s) - z_q^{(i)}\|_2^2$.

The second step ensures that in every $M$ batches all the codes become active again. In Section 4.3, we show that our technique is necessary for training large discrete skill codebooks, as it keeps all codes active, and leads to lower training loss. We provide additional details in Appendix C.

### 3.3 SKILL LEARNING

Choreographer learns a generative world model, which allows the agent to imagine trajectories of future model states $\tau_s = \{s_t, s_{t+1}, ...\}$. A major novelty of our approach is that we learn a skill representation that is linked to model states rather than to environment observations. As a consequence, for skill learning, we define the objective of Choreographer to maximize the mutual information

between trajectories of model states and skills $I(\tau_s, z)$. In visual environments, this is particularly useful as the model states $s_t$ can have a much more compact representation than observations $x_t$.

We rewrite the mutual information objective in forward form as:

$$I(\tau_s, z) = H(\tau_s) - H(\tau_s|z) = H(\tau_s) + \mathbb{E}_{s \sim d_{\pi_z}(s), z \sim \mathrm{Unif}(z)}[\log p(s|z)], \qquad (6)$$

where the second term uses $d_{\pi_z}(s)$ to express the stationary distribution of model states visited by the skill policy conditioned on the skill code $z$, and codes are sampled from a uniform distribution. In order to maximize this objective with RL, we estimate the two terms to be part of an overarching reward function. The entropy term is estimated using a K-NN particle-based estimator (Singh et al., 2003), obtaining $r_{\mathrm{ent}} \propto \sum_{i=1}^{K} \log \|s - s_i^{\mathrm{K\text{-}NN}}\|_2$. For the second term, we can rollout skill-conditioned policies in imagination, obtaining sampled trajectories of model states, and use the distance between the states visited and the decoded state corresponding to the conditioning skill code $z$ to estimate the likelihood (Goodfellow et al., 2016), obtaining $r_{\mathrm{code}} \propto -\|D(z) - s\|_2^2$.

Choreographer learns $N$ code-conditioned skill policies using an actor-critic algorithm:

$$r_{\mathrm{skill}} = r_{\mathrm{ent}} + r_{\mathrm{code}}, \qquad \text{Skill actor:} \quad \pi_{\mathrm{skill}}(a_t|s_t, z), \qquad \text{Skill critic:} \quad v_{\mathrm{skill}}(s_t, z), \qquad (7)$$

where the policies are encouraged to reach states corresponding to their skill-conditioning code $z$, by the $r_{\mathrm{code}}$ term, but at the same time are encouraged to be dynamic and visit different states in one trajectory, by the $r_{\mathrm{ent}}$ term. Given that the reward terms can be computed using the agent's model, the skill actor-critics are trained in the agent's imagination, backpropagating through the model dynamics (Hafner et al., 2021). By doing so, we are able to disentangle exploration and skill learning, as the two processes can happen in parallel or at different times. Visualizations of the skills learned by Choreographer are available in Figure 5 and in the videos on the project website.

### 3.4 SKILL ADAPTATION

Choreographer is able to learn several skill-conditioned policies in a completely unsupervised fashion, leveraging representation and reinforcement learning. This feature is particularly useful during the pre-training stage of unsupervised RL, where the agent learns both a task-agnostic world model of the environment and a set of reusable behaviors. During fine-tuning, the agent needs to leverage such pre-trained components in order to adapt to a downstream task in an efficient way.

Exhaustively deploying the skill policies, a.k.a. the agent's options, in the environment, to understand which ones correlate best with the downstream task, is expensive and often unnecessary. Choreographer can again exploit its world model to deploy the skill policies in imagination.

In order to coordinate the multiple options available, Choreographer learns a *meta-controller*, a policy over options (Precup, 2000) deciding which skill codes use to condition the skill policy:

$$r_{\mathrm{meta}} = r_{\mathrm{task}}, \qquad \text{Meta-ctrl actor:} \quad \pi_{\mathrm{meta}}(z|s_t), \qquad \text{Meta-ctrl critic:} \quad v_{\mathrm{meta}}(s_t). \qquad (8)$$

The meta-controller is updated with rollouts in imagination using the task rewards from a reward predictor, trained only at fine-tuning time. The outputs of the meta-controller's actor network are discrete (skill) vectors and we use policy gradients to update the actor (Williams, 1992). In order to fine-tune the pre-trained skill policies, we can backpropagate the gradients to the skill actors as well (Kingma & Welling, 2013). As we discuss in Section 4.3, this improves performance further as the pre-trained skills are not supposed to perfectly fit the downstream task beforehand.

## 4 EXPERIMENTS

We empirically evaluate Choreographer to answer the following questions:

- **Does Choreographer learn and adapt skills effectively for unsupervised RL?** (Section 4.1) We use the URL benchmark (Laskin et al., 2021) to show that, after pre-training on exploration data, our agent can adapt to several tasks in a data-efficient manner. We show this holds for both training on states and pixels inputs and in both offline and online settings.
- **Do the skills learned by Choreographer provide a more exploratory initialization?** (Section 4.2) We zoom in on the Jaco sparse tasks from URLB and use sparse goal-reaching

tasks from Meta-World (Yu et al., 2019) to evaluate the ability of our agent to find sparse rewards in the environment, by leveraging its skills.

- **Which skills are learned by Choreographer?** (Section 4.3) We visualize the diverse skills learned, highlighting the effectiveness of the proposed resampling technique. Finally, we discuss the 'zero-shot' adaptation performance achievable when fine-tuning only the meta-controller, with no gradients flowing to the skill policies.

Implementation details and hyperparameters are provided in Appendix B and the open-source code is available on the project website.

**Baselines.** For state-based experiments, we adopt the baselines from the CIC paper (Laskin et al., 2022), which can be grouped in three categories: knowledge-based (blue in plots), which are ICM (Pathak et al., 2017), RND (Burda et al., 2019), Disagreement (Pathak et al., 2019); data-based (green in plots), which are APT (Liu & Abbeel, 2021b) and ProtoRL (Yarats et al., 2021); and competence-based (red/orange in plots) APS (Liu & Abbeel, 2021a), SMM (Lee et al., 2019) and DIAYN (Eysenbach et al., 2019). For pixel-based experiments, we compare to the strong baselines from (Rajeswar et al., 2022), combining some of the above approaches with the Dreamer algorithm.

## 4.1 URL BENCHMARK

**Offline data.** Choreographer decouples the exploration and skill learning processes, and can thus learn from offline pre-collected data. To show this, we use the pre-collected datasets of exploratory data released in the ExORL paper (Yarats et al., 2022a). We take the first 2M steps from the datasets corresponding to the state-based URLB domains and use these to pre-train Choreographer offline, learning the world model and the skill-related components. In Figure 3, we compare the fine-tuning performance of our method, leveraging the meta-controller at fine-tuning time, to other methods. In particular, we show the performance when pre-training Choreographer using the data collected by the RND approach (Burda et al., 2019). In Appendix D, we show results with different datasets.

Following (Agarwal et al., 2021), we use the interquartile mean (IQM) and optimality gap metrics, aggregated with stratified bootstrap sampling, as our main evaluation metrics. We find that Choreographer overall performs best on both metrics, particularly outperforming the other approaches on the Walker and Quadruped tasks and performing comparably to CIC and ProtoRL in the Jaco tasks. We highlight that Choreographer performs far better than the other competence-based approaches, SMM, APS, and DIAYN, and also strongly outperforms RND, which is the strategy used to collect its data, showing it's a stronger method to exploit the same exploratory data.

**Parallel exploration.** With no pre-collected data available, Choreographer can learn an exploration policy in parallel to the skills (Mendonca et al., 2021b), to collect data and pre-train its components. This idea is detailed in Appendix A. We use the pixel-based setup of URLB to see how the performance evolves over time as the size of the pre-training data buffer grows.

In Figure 4, we show the performance of Choreographer, collecting data with an exploration policy based on the LBS exploration approach (Mazzaglia et al., 2021) and learning skills simultaneously. The agents are being tested at different times during learning, taking snapshots of the agent at 100k,

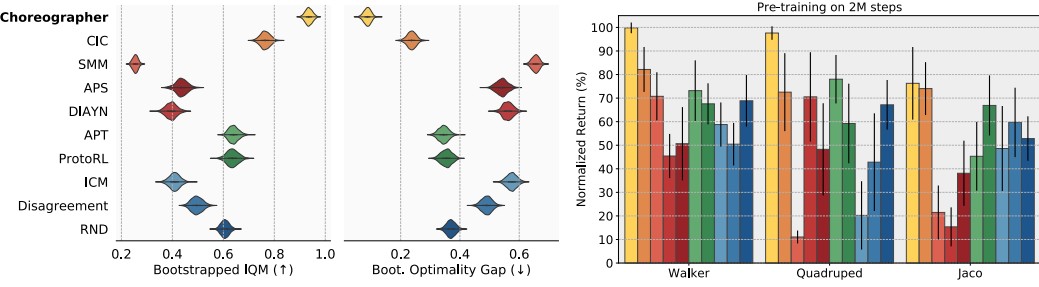

Figure 3: **Offline state-based URLB.** On the left, Choreographer (ours), pre-trained with offline exploratory data, performs best against baselines, both in terms of IQM and Optimality Gap. On the right, mean and standard deviations across the different domains of URLB are detailed (10 seeds).

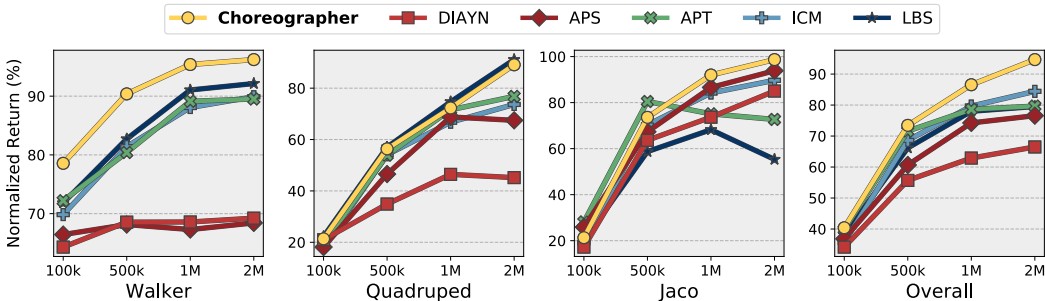

Figure 4: **Pixel-based URLB.** Performance of Choreographer (ours) as a function of pre-training steps. Scores are asymptotically normalized and averaged across tasks for each method. (10 seeds)

500k, 1M, and 2M steps and fine-tuning them for 100k steps. Additional baselines from (Rajeswar et al., 2022), are presented in Appendix D.

Overall, Choreographer's performance is steadily on top, with similar performance to APT, LBS, and ICM until 500k steps, and becoming the one best-performing algorithm from 1M steps on. As also noticed in other works (Laskin et al., 2021), other competence-based approaches APS and DIAYN struggle with the benchmark. Furthermore, both LBS and APT's performance in Jaco starts decaying after 500k steps. As also discussed in (Rajeswar et al., 2022), this is due to the bad initialization of the actor provided by the exploration approach. Our approach overcomes this limitation by learning a set of diverse policies, consistently providing a more general initialization that is easier to adapt.

## 4.2 SPARSE REWARDS

Finding sparse rewards in the environment is challenging, though pre-trained exploratory behaviors may help (Parisi et al., 2021). Choreographer can exploit the skill policies learned during pre-training to reach different model states, efficiently skimming the environment to find rewards. We test this hypothesis using sparse goal-reaching tasks in two pixel-based environments: Jaco from the DMC Suite, and MetaWorld (Yu et al., 2019). In order to solve these tasks, the agent needs to reach within a certain radius of an unknown target goal located in the environment with no feedback.

**Jaco.** We zoom in on the performance on the four reaching tasks from URLB. In Table 1, we show the fraction of runs reaching near the target, without any update of the pre-trained policies, over 100 evaluation episodes. Only Choreographer and random actions found the goals in all runs, the other methods have to rely on entropic exploration to find the goals[1]. The other skill-based methods, APS and DIAYN, also struggled to find the targets consistently.

**Meta-World.** We sampled three sets of 50 goals, in the Sawyer environment from MetaWorld. For each goal in the sets, we evaluate the policies pre-trained by different methods, testing them for 100 evaluation episodes, without any fine-tuning. The performance in Table 1 shows that Choreographer finds the most goals in the environment. All the other approaches struggled in the task, with random actions being the best alternative for higher chances of finding sparse rewards for these goals.

---

[1]For the results in Figure 4 all the baselines (except Choreographer) had an action entropy maximization term in the fine-tuning objective (Haarnoja et al., 2018), helping them to find the goals.

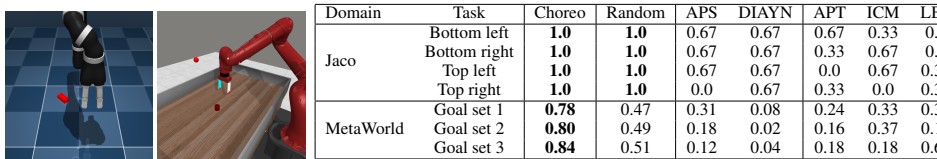

| Domain | Task | Choreo | Random | APS | DIAYN | APT | ICM | LBS | P2E | RND |
|---|---|---|---|---|---|---|---|---|---|---|
| Jaco | Bottom left | **1.0** | **1.0** | 0.67 | 0.67 | 0.67 | 0.33 | 0.0 | 0.67 | 0.67 |
| | Bottom right | **1.0** | **1.0** | 0.67 | 0.67 | 0.33 | 0.67 | 0.0 | 1.0 | 0.33 |
| | Top left | **1.0** | **1.0** | 0.67 | 0.67 | 0.0 | 0.67 | 0.33 | 0.33 | 0.0 |
| | Top right | **1.0** | **1.0** | 0.0 | 0.67 | 0.33 | 0.0 | 0.33 | 0.67 | 0.33 |
| MetaWorld | Goal set 1 | **0.78** | 0.47 | 0.31 | 0.08 | 0.24 | 0.33 | 0.33 | 0.51 | 0.35 |
| | Goal set 2 | **0.80** | 0.49 | 0.18 | 0.02 | 0.16 | 0.37 | 0.18 | 0.41 | 0.43 |
| | Goal set 3 | **0.84** | 0.51 | 0.12 | 0.04 | 0.18 | 0.18 | 0.61 | 0.37 | 0.22 |

Table 1: **Sparse rewards.** Finding goals in the environment with sparse rewards using the pre-trained behaviors. The table shows the fraction of goals found by each method per task. (3 seeds)

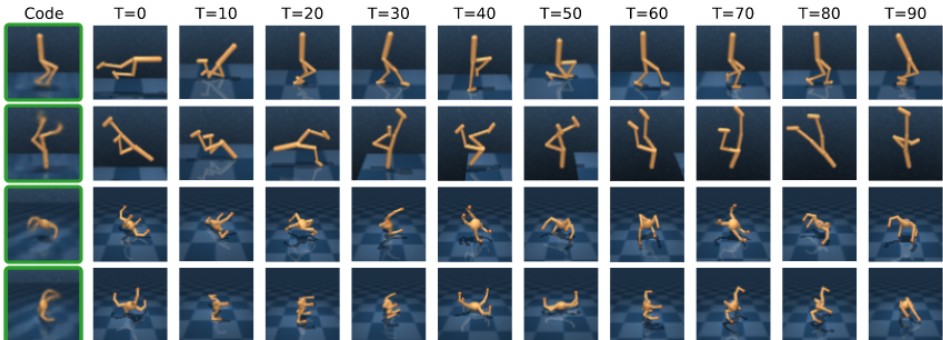

Figure 5: **Skills visualization.** Highlighted in green, reconstructions of the skill codes conditioning the skill behavior on their right, sampling frames over time. Animated results on the project website.

## 4.3 SKILLS ANALYSIS

**Skill visualization.** In Figure 5, we visualize some of the skills learned by Choreographer, in the pixel-based Walker and Quadruped environments. The codes conditioning the skill policies are shown by first reconstructing the corresponding model state, and next the image observation using the reconstruction model. We note that Choreographer's learned behaviors accurately reflect the skill learning objective (Eq. 7): they tend to reach the states corresponding to the conditioning skill codes, but they also dynamically visit different states. For the Walker, this leads to walking-like behaviors for standing-up skill codes (1st row), or behaviors that move around, by hopping or flipping on the torso/legs (2nd row). For the Quadruped, skills tend to learn tipping-over behaviors, where the agent reaches the skill code state alternating rotating and overturning moves (3rd/4th rows). Further visualizations are in Appendix E and on the project website.

**Code resampling.** Learning a diversified set of skill codes is important to learn many useful skill policies for adaptation. Because of index collapse (Kaiser et al., 2018), VQ-VAEs tend to ignore some of the skill codes, choosing not to use them during training. As Choreographer samples skills uniformly for skill learning (Eq. 6), it would be inefficient to train policies that are conditioned on inactive codes. As we show in Figure 7 in the pixel-based URLB environments, code resampling circumvents this issue, making nearly all the codes active during training, reducing the reconstruction loss, and avoiding skill code collapse (see examples of collapsed codes in the Figure).

**Zero-shot skill control.** Pre-training skill policies is a general approach to provide a better initialization of the agent's behavior for fine-tuning to downstream tasks (Eysenbach et al., 2021). As shown in (Eysenbach et al., 2021), it is not possible to learn a set of skill policies able to solve any task, without adaptation. We empirically reinforce this hypothesis with an ablation on the pixel-based URLB, where we learn the meta-controller, which interpolates the skill policies to solve the task, but we do not fine-tune the skill policies. For space constraints, these results, showing a consistent gap in performance versus the skill-adapting counterpart, are presented in Appendix D. Combining multiple skills to solve complex long-term tasks with little to no adaptation, similarly to hierarchical RL (Gehring et al., 2021; Hafner et al., 2022), is a challenge we aim to address in future work.

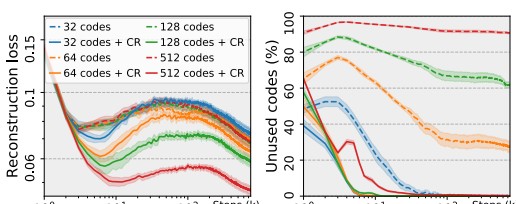

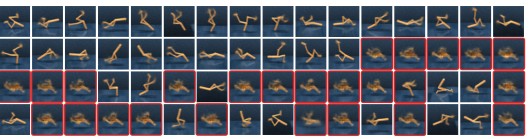

(a) Reconstruction of 64 codes, no code resampling. Inactive, collapsed codes highlighted in red.

Figure 7: **Code resampling (CR).** When using more than 32 codes, CR reduces the number of unused codes ($\sim$0% after 10k steps) and leads to a lower reconstruction loss. (3 seeds $\times$ 3 domains)

## 5 RELATED WORK

**Unsupervised RL.** Among URL methods, we can distinguish (Srinivas & Abbeel, 2021): knowledge-based methods, which aim to visit states of the environment that maximize the agent's model error/uncertainty (Schmidhuber, 1991; Pathak et al., 2017; Burda et al., 2019), data-based methods, which aim to maximize the entropy of the data collected (Hazan et al., 2019; Yarats et al., 2021; Liu & Abbeel, 2021b), competence-based methods, which learn a set of options or skills that allow the agent to control the environment (Gregor et al., 2016; Eysenbach et al., 2019; Hansen et al., 2021). Knowledge and data-based methods focus on expanding the available information, while competence-based methods, such as Choreographer, aim to provide empowerment.

**Exploration and skills.** Exploration is a major challenge in skill learning approaches. Previous work has attempted to solve this issue by using state maximum entropy (Laskin et al., 2022; Liu & Abbeel, 2021a) or introducing variations of the skill discovery objective (Kim et al., 2021; Park et al., 2022). The idea of decoupling exploration and skill discovery has been less explored and mostly applied in simpler environments (Campos et al., 2020; Nieto et al., 2021). To the best of our knowledge, Choreographer represents the first approach that enables scaling the idea to complex environments and tasks successfully from unsupervised data. Additional comparisons in Appendix F.

**Behavior learning in imagination.** The Dreamer algorithm (Hafner et al., 2019; 2021) introduced behavior learning in imagination, establishing a modern implementation of the Dyna framework (Sutton, 1991). The idea has been applied for exploration (Sekar et al., 2020; Rajeswar et al., 2022; Mazzaglia et al., 2021), goal-directed behavior (Hafner et al., 2022; Mendonca et al., 2021b), and planning amortization Xie et al. (2020). In particular, Rajeswar et al. (2022) showed the effectiveness of fine-tuning behavior in imagination on the URL benchmark from pixels, combining unsupervised RL for exploration with Dreamer. Choreographer not only yields better performance but also decouples exploration and behavior pre-training, as the skill policies can be learned offline.

## 6 DISCUSSION

Competence-based approaches aim to learn a set of skills that allow adapting faster to downstream tasks. Previous approaches have struggled to accomplish this goal, mainly due to the impaired exploration capabilities of these algorithms, and the impossibility of efficiently leveraging all the learned skills during adaptation time. In this work, we introduced Choreographer, a model-based agent for unsupervised RL, which is able to discover, learn and adapt skills in imagination, without the need to deploy them in the environment. The approach is thus exploration-agnostic and allows learning from offline data or exploiting strong exploration strategies.

In order to decouple exploration and skill discovery, we learn a codebook of skills, intuitively clustering the agent's model state space into different areas that should be pursued by the skill policies. While this proved to be successful in several domains, it might not be the most effective skill representation, and alternative ways for learning the skills could be explored, such as identifying crucial states in the environment (Faccio et al., 2022).

Choreographer can evaluate skill policies in imagination during the fine-tuning stage and use a meta-controller to decide which policies should be employed and adapted to the downstream task. While our method exploits a world model for data-efficient adaptation, alternative strategies such as successor representations (Hansen et al., 2020) or meta RL (Zintgraf et al., 2019) could be considered.

The meta-controller allows evaluating all the skill policies in imagination and fine-tuning them for downstream tasks efficiently, but currently doesn't allow combining the skills for longer-horizon complex tasks. We aim to further investigate this in the future, leveraging hierarchical RL (Hafner et al., 2022) and model learning, as established in the options framework (Precup, 2000).

**Limitations.** Choreographer is a data-driven approach to skill discovery and learning. The skill representation depends on the data provided or collected from the environment and on the world model learned. As a consequence, the skills learned by Choreographer tend to reach multiple areas of the environment and assume different behaviors, but struggle to capture the naturalness that humans develop in their skills. We leave the idea of incorporating human bias in the skill learning process for future work.

**Reproducibility statement**   We reported in appendix the detailed pseudo-code for both Choreographer (Algorithm 2) and code-resampling (Algorithm 1). All the instructions and training hyperparameters to implement and reproduce the results are in Appendix B, along with a summary of other experiments we tried that could be useful for future work. The code is publicly available through the project website.

## ACKNOWLEDGEMENTS

This research received funding from the Flemish Government (AI Research Program). Pietro Mazzaglia is funded by a Ph.D. grant of the Flanders Research Foundation (FWO).

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

## APPENDIX

## A    DATA COLLECTION

In previous skill discovery methods (Eysenbach et al., 2019; Liu & Abbeel, 2021a; Achiam et al., 2018), skills are primarily used as a means to discover the environment by maximizing the diversity of skills. In contrast, Choreographer's skill discovery is exploration-agnostic, in that it discovers and learns skills using its model. For this reason, Choreographer can be adopted in two ways:

- using a pre-collected dataset used to learn the world model and skill-related components;
- using an exploration policy, which aims to collect diverse data to pre-train.

When not using pre-collected data, we learn an exploration actor-critic in imagination that maximizes intrinsic rewards (Mendonca et al., 2021b; Sekar et al., 2020). In particular, we shape rewards following LBS (Mazzaglia et al., 2021), which has proven to be a reliable exploration method when using world models (Rajeswar et al., 2022). Exploration rewards approximate the information gain between observations and model states, computed by measuring the KL divergence between the posterior and the prior of the model. The exploration actor-critic is defined as follows:

$$\text{Expl. actor:} \quad \pi_{\text{expl}}(a_t|s_t), \qquad \text{Expl. critic:} \quad v_{\text{expl}}(s_t), \tag{9}$$

$$r_{\text{expl}} = D_{\text{KL}}[q_\phi(s_t|s_{t-1}, a_{t-1}, x_t) \| p_\phi(s_t|s_{t-1}, a_{t-1})],$$

## B    IMPLEMENTATION

### B.1    AGENT

**World Model.** We follow the architecture of the DreamerV2 agent's world model (Hafner et al., 2021). Model states have both a deterministic and a stochastic component: the deterministic component is the 200-dimensional output of a GRU (Cho et al., 2014), with a 200-dimensional hidden layer; the stochastic component consists of 32 categorical distributions with 32 classes each. For states-input experiments, the encoder and decoder are 4-layers MLP with a dimensionality of 400. For pixels-based experiments, the encoder and decoder follow the architecture of DreamerV2, taking $64 \times 64$ RGB images as inputs.

**Skill auto-encoder.** The encoder and decoder networks are 4-layer MLP with a dimensionality of 400. The input and target of the auto-encoder are the deterministic part of the world model states. The codebook consists of 64 codes of dimensionality 16. The gradients of the skill auto-encoder objective are back-propagated through the decoder and the encoder using straight-through gradients (Bengio et al., 2013). The codes of the codebooks are updated using an exponential moving average of their assigned embeddings (Razavi et al., 2019). Code resampling happens every $M = 200$ training batches.

Both the world model and the skill-autoencoder networks are updated sampling batches of 50 sequences of 50 timesteps, using Adam with learning rate $3 \cdot 10^{-4}$ for the updates, and clipping gradients norm to 100.

**Skill actor-critic.** Skill actor-critic takes as input both model states and skill vectors (Schaul et al., 2015). We use the Dreamer algorithm to update them in latent imagination (Hafner et al., 2019), imagining sequences of 15 steps with the model, starting from the world model training batch states. The actor and critic networks are instantiated as 4-layer MLP with a dimensionality of 400 and updated using Adam with a learning rate of $8 \cdot 10^{-5}$ and gradients clipped to 100. The actor's output is a truncated normal distribution and its gradients are backpropagated using the reparameterization trick Kingma & Welling (2013).

Rewards are computed in latent imagination using:

$$r_{\text{skill}}(s, z) = \underbrace{\frac{1}{K} \sum_{i=1}^{K} \|s - s_i^{\text{K-NN}}\|_2}_{r_{\text{entropy}}} - \underbrace{\|D(z) - s\|_2}_{r_{\text{code}}}.$$

For the entropy term, where we approximate a particle-based entropy estimator (Singh et al., 2003), we follow Liu & Abbeel (2021b) in averaging the distance from the K-nearest neighbors ($K = 30$) and remove the $\log$. For the code term, we take the L2-norm directly instead of the squared L2-norm. With these minor modifications, the two terms are on the same scale and we found it unnecessary to add any scaling coefficient to balance the two reward terms.

**Meta-controller actor-critic.** The meta-controller actor-critic is instantiated and trained in the same way as the skill actor-critic, using the environment rewards provided by the model's reward predictor. While the meta-controller is updated using policy gradients (Williams, 1992), we can backpropagate the meta-controller loss to the skill actor-critic to fine-tune them, with the reparameterization trick.

One issue we found is that when the rewards are too small (or zero) to predict, the predictions of the reward predictor and the critic are noisy around zero, and tend to collapse the skill behaviors during fine-tuning. In order to avoid that, we introduce a reward smoothing mechanism, which makes sure the critic and reward predictions are exactly zero, until a reward $\geq 1 \cdot 10^{-4}$ is found. By doing so, we ensure that, until an actual reward is encountered, the skill policies remain the ones learned during pre-training and can be used to explore the environment, looking for rewards. To provide consistent exploration, skills are sampled and followed for an entire trajectory, until rewards are encountered.

## B.2 EXPERIMENTS

**URLB.** The URL benchmark (Laskin et al., 2021) is designed to compare different URL strategies. The experimental setup consists of two phases: a longer data collection/pre-training phase, where the agent has up to 2M environment steps to interact with the environment without rewards, and a shorter fine-tuning phase, where the agent has 100k steps to interact with a task-specific version of the environment, where it should both discover rewards and solve a downstream task. In the URLB, there are three domains from the DM Control Suite (Tassa et al., 2018), Walker, Quadruped, and Jaco, with four downstream tasks to solve per each domain.

For state-based inputs, we used an action repeat of 1. For pixel-based inputs, we used an action repeat of 2. For the state-based experiments, we note that the Jaco environment in ExORL slightly differs from the one in URLB, as the position of the target is not provided to the agent in ExORL. Other than that, the Jaco tasks remain the same and we see that, nonetheless, Choreographer is able to find the targets without information about their position, performing comparably to the top-performing baselines.

In both URLB settings, all the agent's components are updated once every 10 actions (5 steps, when considering action repeat of 2). This is in line with the baselines from (Rajeswar et al., 2022). For pre-training, this amounts to a total of 200k update steps in both settings, which are performed using a fixed dataset in the offline URLB from states experiments, and with a replay buffer growing over time, in the parallel exploration URLB from pixels experiments. The same update frequency is used during the fine-tuning stage.

For the sparse-rewards Jaco, we consider the target reached when the agent finds a reward $\geq 1 \cdot 10^{-4}$.

**MetaWorld.** In MetaWorld, the task can be adapted to be sparse, by setting rewards equal to the success metrics in the environment. For the `reach` tasks, success is when the agent reaches the target. In order to build the three goal sets, we adapted the code to sample 50 goals in the intervals: $\text{goal\_low} = (-0.45, 0.3, 0.1), \text{goal\_high} = (0.45, 0.9, 0.5)$, with seeds $(1, 2, 3)$.

For both sparse experiments, the agents are pre-trained on 2M frames collected using their exploration policy (Choreographer uses LBS as detailed in Appendix A). Then, the pre-trained behavior is deployed for evaluation, without any information about the goals to find. For the skill-based approaches, which are DIAYN, APS and Choreographer, skills are randomly re-sampled every 50 steps.

## B.3 OTHER THINGS WE TRIED

**Skill Discovery.** We tried alternative representations other than the skill auto-encoder, for skill discovery. In particular, we tried contrastive learning and soft clustering using the Sinkhorn-Knopp algorithm (Yarats et al., 2021). While both of them worked, we found the skill vectors found by the autoencoder to be more diverse and explainable (thanks to the reconstruction). We also tried: (i)

using a standard discrete VAE, using straight-through gradients (Bengio et al., 2013), compared to a VQ-VAE, but the training was more unstable; (ii) using a VAE with a gaussian parameterization, discovering skills in a continuous latent space, but we found this configuration to perform worse at adaptation time, as exploring the environment using the large skill space and adapting a potentially infinite number of skills with the meta-controller used to require more data/training.

**Skill learning.** We attempted to learn different numbers of skills, ranging from 16 up to 512. We found that the best performance was obtained when using a number of skills comprised between 64 and 128, and we stick to 64 for our experiments. Too few skills were sometimes unable to find useful behaviors in the Quadruped or to find sparse targets in the Jaco tasks. Too many skills, under the same computational budget, generally led to skills that didn't fully converge or were harder to fine-tune. Though, likely, with a higher computational budget, more skills might perform better.

**Skill adaptation.** The meta-controller generally samples skills at a time-frequency of 1. Other works have focused on learning skills that last several timesteps (Sharma et al., 2020), using them for hierarchical control (Gehring et al., 2021). Choreographer, instead, tends to evaluate all skills in imagination and narrow down its choice to a restricted set of skills that are adapted for the downstream tasks. We found that sampling skills with lower frequency (we tried every 3,5,15,50 steps) does not lead to better adaptation performance, but often does not harm performance either. We believe that leveraging the skills for hierarchical problems is an interesting direction for future work.

### B.4 Considerations about efficiency

**Data efficiency.** Choreographer is a method that allows for greater data-efficiency, after pre-training a model and skills from unsupervised data. We show that the agent achieves state-of-the-art performance when fine-tuned for 100k steps in the URLB settings, both from pixels and states input.

**Parameter efficiency.** Choreographer is a model-based agent, training a world model of $\sim$20M parameters for pixel-based environments and $\sim$8M parameters for state-based environments. For the skill components, a skill autoencoder of $\sim$1M parameters and skill policies requiring $\sim$2M parameters for their actor-critic are trained. During fine-tuning, the meta-controller requires an additional 2M parameters for its actor-critic.

Other skill learning approaches tend to use a similar magnitude of parameters for their skill components, e.g. APS and DIAYN require $\sim$1M parameters each for their networks, CIC requires 4M parameters. When these techniques are used in a model-free fashion, as the state-based URLB baselines (Laskin et al., 2021; 2022), Choreographer number of parameters can be slightly higher because of the world model and the meta-controller [2]. When these techniques are used in a model-based fashion, as the pixel-based URLB baselines (Rajeswar et al., 2022), the only additional parameters are for the meta-controller.

**Computation efficiency.** In the URLB paper, they claim a training time of about 12 hours for 2M frames pre-training using V100 GPUs for their model-free agents. Using the same GPUs, we can pre-train a Choreographer agent in about 44 hours. The main slowdowns are due to: (i) the GRU network in the world model, which needs to process sequences of inputs sequentially, both during model training and in the imagination process, and (ii) the imagination process, which is not executed in parallel with the model training.

## C Code Resampling

In RL, we mostly face datasets that grow arbitrarily over time, as the agent collects new data. These datasets may become large (in our case, up to $2 \cdot 10^6$ environment transitions) and thus require a long time to run clustering algorithms, such as k-means. Furthermore, as the data increases, the algorithm may need to be run again, several times, to comply with the new data.

By using a VQ-VAE, we are able to perform clustering of the model state space, as well as to decode the cluster centers into their corresponding model states (which is necessary for Choreographer skill learning objective). However, VQ-VAE suffers from index collapse.

---

[2] As a quick overview, the model-free DrQ-v2 agent (Yarats et al., 2022b) requires about 7M parameters for pixel-based inputs and DDPG (Lillicrap et al., 2016) requires 3M parameters for state-based inputs.

We propose a code resampling technique that is inspired by the k-means++ initialization (Arthur & Vassilvitskii, 2006). This technique is generally applied with fixed datasets to reduce the k-means convergence time, by providing a better initialization. Instead, we use the code re-initialization procedure to 'resample' codes every $M$ training batches, making sure that any collapsed index/code is reactivated.

Our technique scales well for the RL setting, as it uses only the data in the previous batch to resample the code. Of course, this may lead to a suboptimal code choice, which can then be adjusted at following code resampling iterations, if necessary. We showed in Figure 7 that in practice all the codes tend to stay active over training.

---

**Algorithm 1:** Code Resampling

```
if train_step mod M == 0:
    for code in inactive_codes:
        distance_matrix = l2norm_distance(current_batch, current_codes)
        probabilities = min(distance_matrix, axis=code_axis) \
                        / sum(min(distance_matrix, axis=code_axis))
        code = current_batch[sample_index(probs=probabilities)]
```

---

In Figure 8, we investigate the robustness of our code resampling technique to the hyperparameter $M$, i.e. "resample every $M$ train steps". In the experiment, we use a codebook dimensionality of 512 and train the agent for 2M steps. We found that the reconstruction loss eventually converges to nearly the same quantity for different values of $M$. However, using a value of $M$ of 10 or 100 leads to a lower loss earlier and to a lower percentage of unused codes, compared to using larger values (1000 or 10000). We also experimented with a value of $M = 1$ but we find instabilities during training, because of the excessive resampling. Overall, the technique is robust to a large range of hyperparameters, but we suggest setting $M$ to lower values if earlier convergence can be useful. For Choreographer, we used 200 as a default hyperparameter.

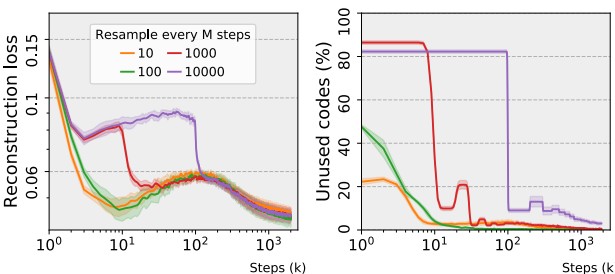

Figure 8: **Code resampling hyperparameter robustness.** The plots show the reconstruction loss and the percentage of unused codes, whenvarying the hyperparameter $M$ (3 seeds × 3 domains).

## D    ADDITIONAL RESULTS

**URLB from states.** In Table 2, we present detailed results and baselines on the state-based URLB. Choreographer performs best on all the Walker and Quadruped tasks and comparably to CIC in the Jaco tasks. The normalization scores for the results in the paper are the Expert results.

In Figure 3, we showed the results of Choreographer pre-trained with 2M steps of exploratory data from the RND approach. In Figure 9, we compare to the results when pre-training with data collected by other knowledge-based approaches, i.e. ICM, Disagreement, or random actions data. We see that the quality of the model and the skill learned, and so the fine-tuning performance depends on the data used. Choreographer is able to learn skills that widely explore the model state space. However, if the state space explored is little, then the skills will be also limited to a restricted area. This is what generally happens for the case of the random actions in the Figure, but also in the Quadruped

| Pre-trainining for $2 \times 10^6$ environment steps | | | | | | | | | | | | |
|---|---|---|---|---|---|---|---|---|---|---|---|---|
| Domain | Task | Expert | DDPG | ICM | Disagreement | RND | APT | ProtoRL | SMM | DIAYN | APS | CIC | Choreographer |
| Walker | Flip | 799 | 538±27 | 417±16 | 346±13 | 474±39 | 544±14 | 456±12 | 450±24 | 319±17 | 465±20 | 631±34 | **960 ± 3** |
| | Run | 796 | 325±25 | 247±21 | 208±15 | 406±30 | 392±26 | 306±13 | 426±26 | 158±8 | 134±16 | 486±25 | **635 ± 10** |
| | Stand | 984 | 899±23 | 859±23 | 746±34 | 911±5 | 942±6 | 917±27 | 924±12 | 695±46 | 721±44 | 959±2 | **980 ± 1** |
| | Walk | 971 | 748±47 | 627±42 | 549±37 | 704±30 | 773±70 | 792±41 | 770±44 | 498±27 | 527±79 | 885±28 | **967 ± 1** |
| Quadruped | Jump | 888 | 236±48 | 178±35 | 389±62 | 637±12 | 648±18 | 617±44 | 96±7 | 660±43 | 463±51 | 595±42 | **830 ± 9** |
| | Run | 888 | 157±31 | 110±18 | 337±30 | 459±6 | 492±14 | 373±33 | 96±6 | 433±29 | 281±17 | 505±47 | **779 ± 11** |
| | Stand | 920 | 392±73 | 312±68 | 512±89 | 766±43 | 872±23 | 716±56 | 123±11 | 851±43 | 542±53 | 761±54 | **946 ± 5** |
| | Walk | 866 | 229±57 | 126±27 | 293±37 | 536±39 | 770±47 | 412±54 | 80±6 | 576±81 | 436±79 | 723±43 | **921 ± 5** |
| Jaco | Reach bottom left | 193 | 72±22 | 111±11 | 124±7 | 110±5 | 103±8 | 129±8 | 45±7 | 39±6 | 76±8 | 138±9 | 137 ± 10 |
| | Reach bottom right | 203 | 117±18 | 97±9 | 115±10 | 117±7 | 100±6 | 132±8 | 46±11 | 38±5 | 88±11 | 145±7 | 147 ± 10 |
| | Reach top left | 191 | 116±22 | 82±14 | 106±12 | 99±6 | 73±12 | 123±9 | 36±3 | 19±4 | 68±6 | 153±7 | 164 ± 7 |
| | Reach top right | 223 | 94±18 | 103±11 | 139±7 | 100±6 | 90±10 | 159±7 | 47±6 | 28±6 | 76±10 | 163±4 | 169 ± 10 |

Table 2: Mean and standard error performance of Choreographer on state-based URLB after pre-training for 2M steps and fine-tuning 100k steps. Baseline scores from (Laskin et al., 2022).

for ICM (where we visually inspected the data from ExORL, verifying that ICM explores poorly in the Quadruped environment).

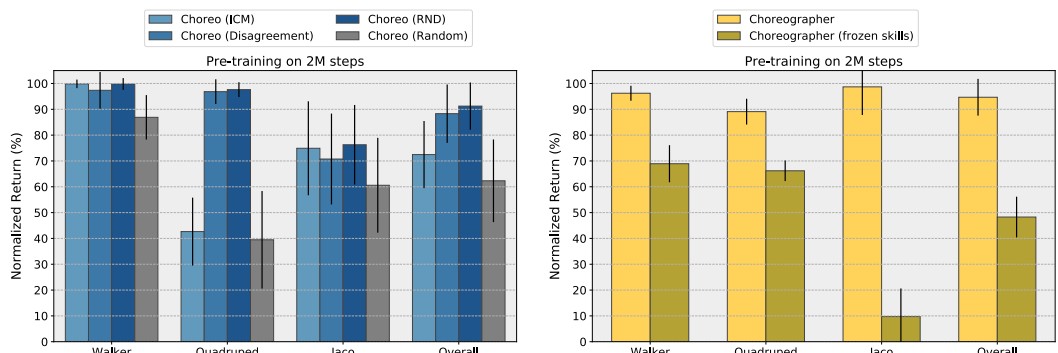

Figure 9: **Different data.** Training Choreographer's world model and skills using data from different exploration strategies.

Figure 10: **Zero-shot skill control.** We attempt to solve the task using the pre-trained skill policies, without any fine-tuning.

In Figure 11, we present an ablation analysis of our approach when removing the meta-controller, and instead adapting only one of the skills. The skill adapted is selected to be the one with the highest expected rewards, considering the states and rewards obtained in the initial 4k steps of the fine-tuning stage. We see the performance remains unchanged in the Walker and Quadruped tasks, where the agent mostly requires a good initialization to master the downstream task quickly. In the Jaco tasks, instead, we see that having a meta-controller that considers all the skills is still beneficial. We believe this is because the task is sparser and maintaining more skills available helps explore the environment better during fine-tuning. We also highlight that the performance of Choreographer without the meta-controller is overall higher than the second-best approach (CIC).

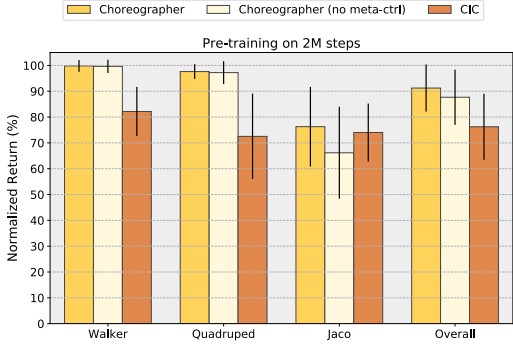

Figure 11: **Offline URLB ablation.** Ablations on the meta-controller for the state-based URLB experiments.

**URLB from pixels.** In Figure 4, we showed the performance of Choreographer in the pixel-based setting over time. This performance was achieved by learning the meta-controller and adapting the skill policies during fine-tuning. In Figure 10, we show the impact of fine-tuning the skill policies learned during pre-training, after 2M steps. This can be seen as restricting the agent's action space to a set of N state-dependent actions (=64 skills in this case), which the agent can use to solve the downstream task.

The performance on the Walker and Quadruped tasks is weaker, but the agent still performs comparably to some weaker baselines in Figure 4, such as APS and DIAYN. Instead, the performance on Jaco is much worse than the one with skill adaptation, that's because the task requires precision, to reach the target block, and adaptability, in case the block moves.

In Figure 12, we present two ablation analyses of our approach: removing code resampling, so that some of the skills trained are meaningless because of index/code collapse, and removing the meta-controller, adapting instead only one of the skills, selected after the initial 4k frames of the fine-tuning stage. We find that Choreographer always improves performance with respect to its ablations. In particular, both using the meta-controller and making sure that all the skills are active and properly learned, seem to be useful in the Walker tasks, especially in the first 500k steps, and in the Jaco tasks, at all fine-tuning times.

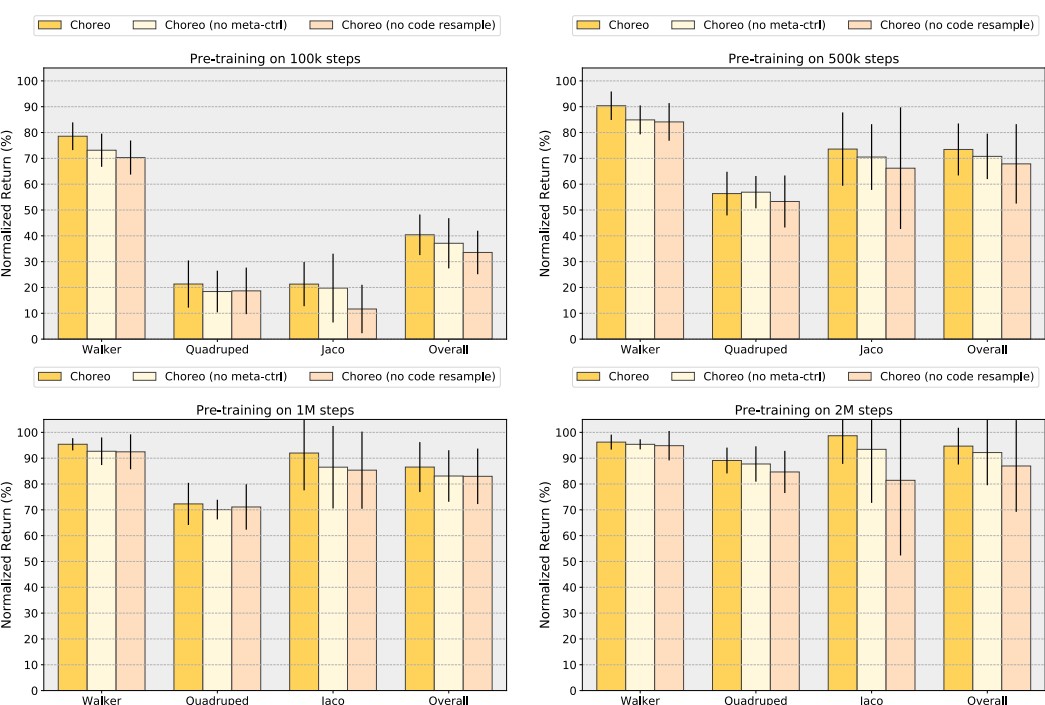

Figure 12: **Pixel-based URLB ablations.** Ablations on the meta-controller and code resampling for the pixel-based URLB experiments.

In Figure 13, we show additional results, including more baselines from (Rajeswar et al., 2022), on the pixel-based URLB. Overall, Choreographer clearly outperforms all methods, while LBS is still competitive on the Quadruped.

**Comparison with SPIRL.** As we discuss in Appendix F, previous work on offline skill learning also showed promising performance. However, these methods generally employ demonstrations or play data collected by humans. In contrast, Choreographer is able to leverage vast amounts of unsupervised data. To strengthen this thesis we compare with the SPIRL approach (Pertsch et al., 2020) on the state-based URLB. SPIRL learns different skill policies from offline data autoencoding trajectories in the given dataset, i.e. learning to imitate trajectories in the data through the skills. Then, during adaptation, the skills are combined using a high-level policy that decides which skill should be applied for a fixed time horizon. This policy is also regularized by a high-level skill prior

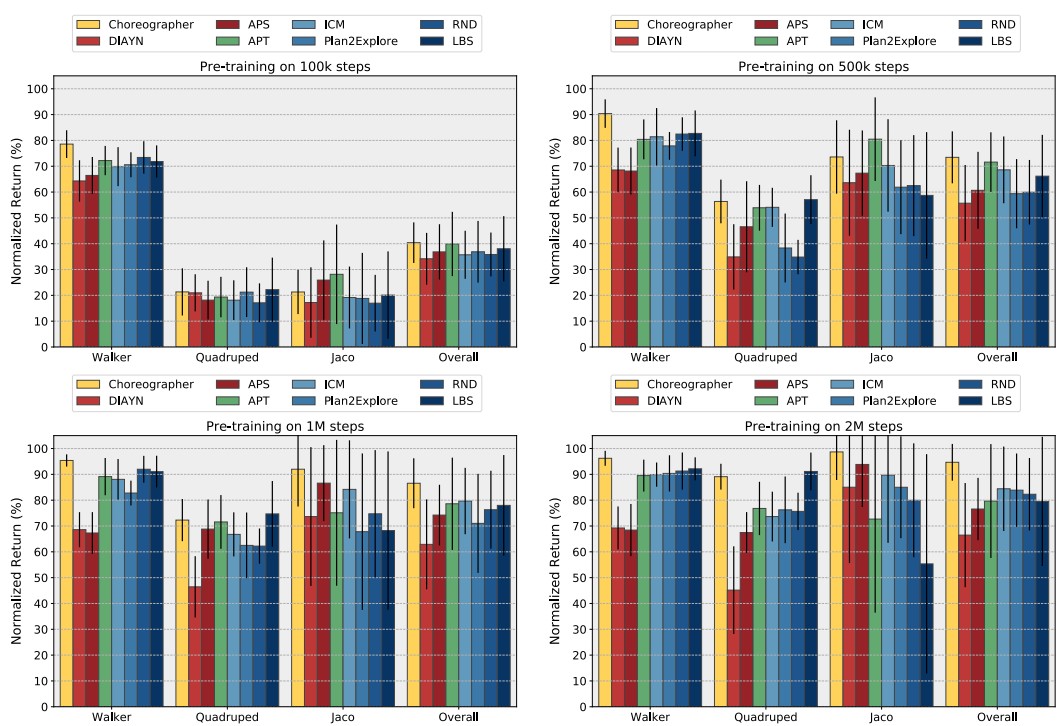

Figure 13: **Pixel-based URLB.** Additional baselines for the pixel-based URLB experiments.

that is learned on the offline dataset. In Figure 14, we report results obtained using the original paper hyperparameters. SPIRL struggles to solve all the tasks in URLB leveraging its learned skill action space. We believe the main reason is that skills learned by imitating unsupervised exploration trajectories are generally less useful for solving downstream tasks. We experimented with different hyperparameters, such as varying the skill horizon, reducing the constraint w.r.t. the skill prior, and increasing the capacity of the neural networks used. While these choices led to some improvements in performance, proving that both skill priors and long-term skills are less successful when employing unstructured data, the performance of SPIRL remained underwhelming in this unsupervised offline setting.

**Comparison with model-based EDL.** As we detail in Appendix F, the Explore Discover and Learn (EDL; (Campos et al., 2020)) shares some similarities with Choreographer. In order to show the relevance of the differences between Choreographer and a combination of EDL + Dreamer, we run additional experiments on URLB. In these experiments, EDL + Dreamer can be considered an ablation of Choreographer where we: (i) use EDL's objective instead of Choreographer's to learn the skills in imagination, (ii) remove code resampling, (iii) remove the meta-controller. Noting that EDL uses the mutual information between skills and environment states, compared to Choreographer which uses the mutual information between skills and latent model states, we have also experimented with a Latent EDL configuration, which uses EDL's objective but with latent states rather than environment states. Compared to Choreographer, Latent EDL still uses no entropy term in the skill reward, no meta-controller and no code resampling. Both in the state-based (Figure 15) and in the pixel-based (Figure 16) the performance of EDL is inferior to Latent EDL, showing that using latent codes is beneficial and Latent EDL is inferior to Choreographer, showing the usefulness of code resampling and of the meta-controller.

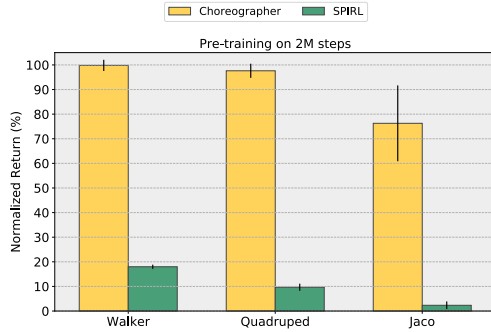

Figure 14: **State-based comparison with SPIRL.** Comparison with SPIRL (Pertsch et al., 2020).

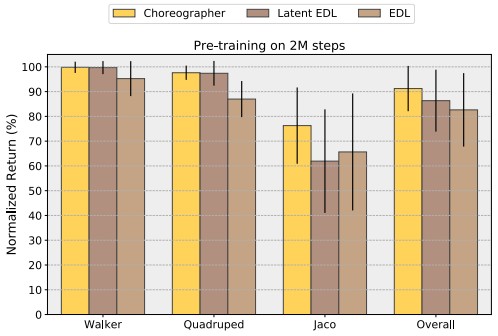

Figure 15: **State-based comparison with EDL.** Comparison with different combinations of EDL (Campos et al., 2020) with Dreamer.

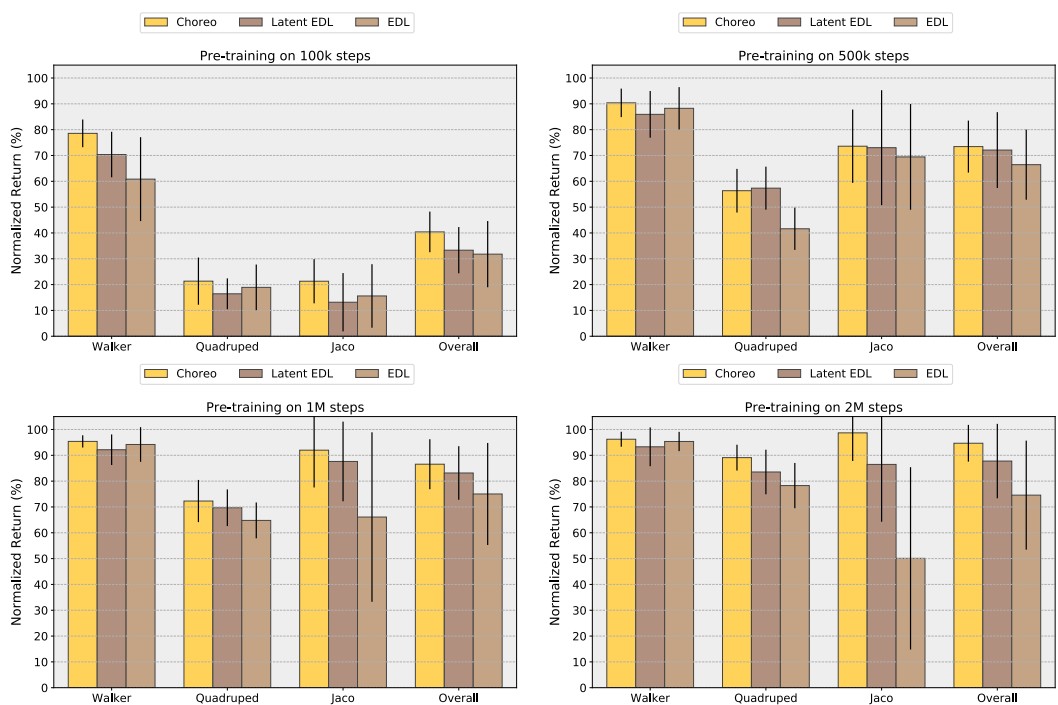

Figure 16: **Pixel-based comparison with EDL.** Comparison with different combinations of EDL (Campos et al., 2020) with the Dreamer algorithm.

# E    ADDITIONAL VISUALIZATIONS

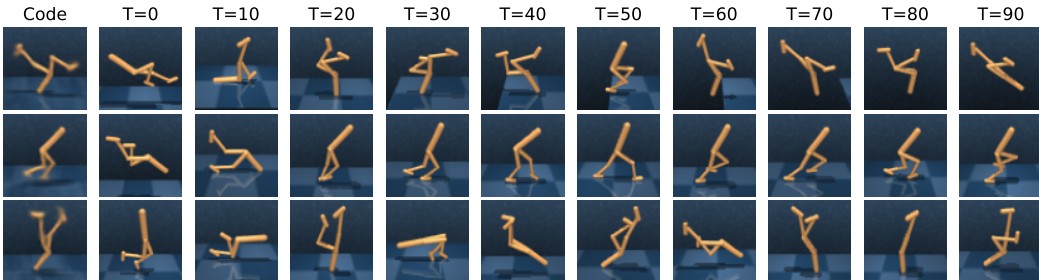

Figure 17: **Walker.** Sets of three skill codes and respective behaviors visualized for Walker.

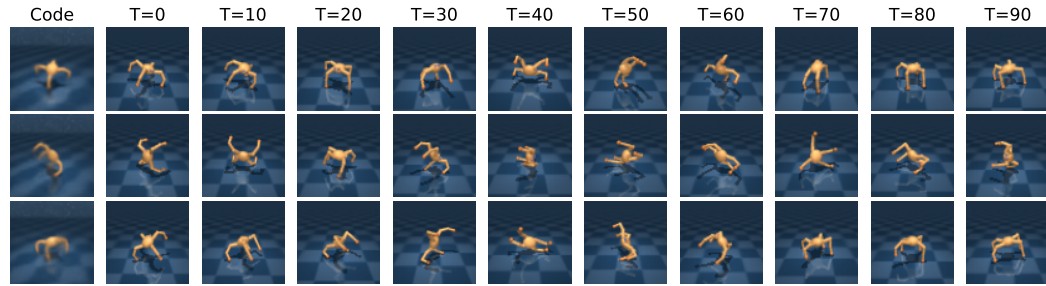

Figure 18: **Quadruped.** Sets of three skill codes and respective behaviors visualized for Quadruped.

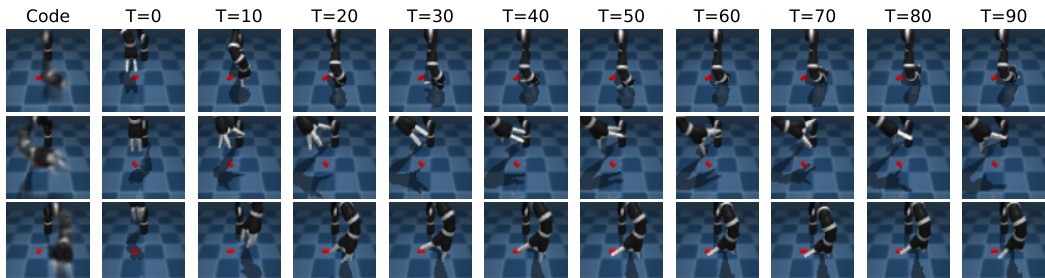

Figure 19: **Jaco.** Sets of three skill codes and respective behaviors visualized for Jaco.

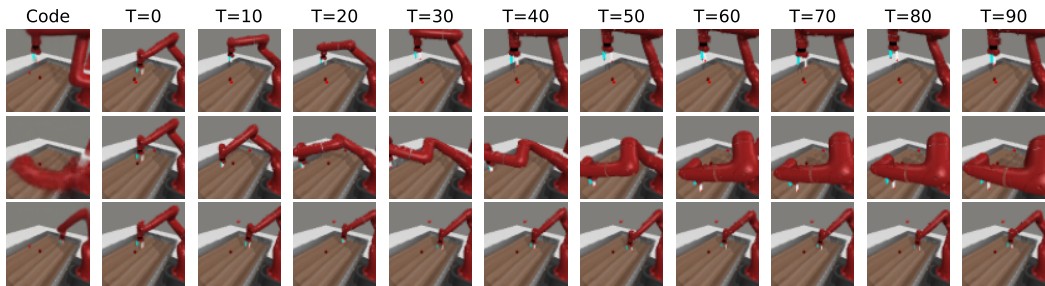

Figure 20: **MetaWorld.** Sets of three skill codes and respective behaviors visualized for MetaWorld.

## F  ADDITIONAL RELATED WORK

**Offline Skill Learning.** Choreographer, by decoupling exploration and skill learning, allows learning skills from offline data that is collected in a completely unsupervised manner. Previous work has also shown that skill learning from offline data is possible and can be especially useful to quickly adapt to complex long-term tasks (Springenberg et al., 2018; Shankar et al., 2020). The skill representation can be used to compress sequences of meaningful actions (Pertsch et al., 2020; Singh et al., 2020; Ajay et al., 2020) or to reach designed goals (Gupta et al., 2019; Lynch et al., 2019). One problem with these approaches is that they generally require the training dataset to contain sequences of coherent actions, e.g. including demonstrations (Pertsch et al., 2021; Kipf et al., 2018), human play data (Gupta et al., 2019; Lynch et al., 2019), or incorporating human feedback in the learning process (Wang et al., 2021). Choreographer, in contrast, makes no assumption about the training dataset: it can extract a set of skills in a completely unsupervised manner and learn meaningful behaviors to accomplish them in imagination, using the model.

**Detailed comparison with EDL.** Explore Discover and Learn (EDL; (Campos et al., 2020)) is an unsupervised skill approach that is related to Choreographer, as both methods decouple exploration and skill learning by using a VQ-VAE to discover skills from data. However, several differences are present between the two methods: (i) Choreographer is a model-based agent, where the model is used to discover, learn and adapt skills, while EDL does not include a model, (ii) Choreographer's skill learning objective is derived from the mutual information between latent model states and skills, while EDL's objective is derived from the mutual information between environment states and skills, (iii) Choreographer's skill learning reward is made of a code-achieving component and of an entropy term, that is maximized in the imagined trajectories, while EDL has no entropy term as the entropy is considered fixed in the offline dataset trajectories, (iv) Choreographer adopts a meta-controller during fine-tuning, which allows the agent to evaluate and fine-tune multiple skills at once.

**Detailed comparison with LEXA.** Latent Explorer Achiever (Mendonca et al., 2021a) is an approach that combines an exploration policy and a goal achieving policy, both learned in imagination, to reach complex goals in the environment. There are some similarities between Choreographer and LEXA, as they are both model-based agents that learn skill/goal-conditioned policies in imagination, without interfering with the exploration process. There are also several differences that can be summarized as follows: (i) Choreographer is designed to work in a setting where, after an unsupervised pre-training phase, the agent is given a short fine-tuning phase to adapt for a given task, by maximizing rewards, while LEXA is designed to accomplish visual goals in a zero-shot setting, after interacting with the environment, (ii) Choreographer learns a discrete latent representation and sample codes internally to learn skill policies, while LEXA samples goal images from an external replay buffer and encodes them as targets for the achiever policies, (iii) Choreographer's objective is derived from a mutual information term and is made of an entropy term and a code-achieving term, while LEXA uses either cosine distance in image embedding space or temporal distance between encoded goals and imagined states, (iv) Choreographer adopts a meta-controller during fine-tuning, which allows the agent to evaluate and fine-tune multiple skills at once.

# G  ALGORITHM

---

**Algorithm 2:** Choreographer

---

Initialize agent modules: world model, skill autoencoder, skill policies

**if** *no replay buffer available* **then**

    Initialize replay buffer.

    Initialize exploration policy.

    *should_explore* = True.

**end**

**while** *is unsupervised phase* **do**

    // Data collection

    **if** *should_explore* **then**

        Update model state with the posterior $s_t \sim q(s_t|s_{t-1}, a_{t-1}, x_t)$.

        Sample exploratory action $a_t \sim \pi_{\text{expl}}(a_t|s_t)$.

        Collect new observation from the environment $x_{t+1}$

        Add $(x_t, a_t, x_{t+1})$ to the replay buffer.

    **end**

    // Pre-training

    **if** $t \bmod train\_every\_n\_steps = 0$ **then**

        Draw a batch of past trajectories $\{x, a\}$ from the replay buffer.

        Update world model on batch (Eq. 3) and get states $\{s\}$.

        **if** should_explore **then**

            Imagine exploration trajectories $\{s, a, z\}$, sampling actions from $\pi_{\text{expl}}(a_t|s_t)$.

            Compute exploration rewards $r_{\text{expl}}$ (Eq. 9).

            Update exploration actor-critic.

        **end**

        // Skill discovery

        Update skill autoencoder using the model states $\{s\}$ (Eq. 5).

        // Skill learning

        Sample skill codes from uniform distribution $z \sim p(z)$.

        Imagine skill trajectories $\{s, a, z\}$, sampling actions from $\pi_{\text{skill}}(a_t|s_t, z)$.

        Compute skill rewards $r_{\text{skill}}$ (Eq. 7).

        Update skill actor-critic.

    **end**

**end**

// Fine-tuning

Reset replay buffer and initialize meta-controller.

**while** *is supervised phase* **do**

    Update model state with the posterior $s_t \sim q(s_t|s_{t-1}, a_{t-1}, x_t)$.

    Sample skill $z \sim \pi_{\text{meta}}(z|s_t)$ and action $a_t \sim \pi_{\text{skill}}(a_t|s_t, z)$.

    Collect new task observation from the environment $x_{t+1}, r_{t+1}$

    Add $(x_t, a_t, x_{t+1}, r_{t+1})$ to the replay buffer.

    **if** $t \bmod train\_every\_n\_steps = 0$ **then**

        Draw batch of past trajectories $\{x, a, r\}$ from the replay buffer.

        Update world model on batch (Eq. 3) and get states $\{s\}$.

        // Skill adaptation

        Imagine traj. $\{s, a, z\}$, sampling skills from $\pi_{\text{meta}}(z|s_t)$ and actions from $\pi_{\text{skill}}(a_t|s_t, z)$.

        Predict task rewards $r_{\text{task}}$

        Update meta-controller actor-critic and propagate gradients to skill actors.

    **end**

**end**

---

