# OpenReview forum: "Choreographer: Learning and Adapting Skills in Imagination"
_ICLR.cc/2023/Conference — ICLR 2023 notable top 25%_

### Official Review · Reviewer_2p4W · 2022-10-21

**Confidence:** 4
**Correctness:** 4
**Technical Novelty And Significance:** 2
**Empirical Novelty And Significance:** Not applicable
**Recommendation:** 6

**Clarity, Quality, Novelty And Reproducibility:**

Clarity/Quality: The paper is well-written and easy to follow.

Novelty: The novelty of the paper is somewhat limited (please see above).

Reproducibility: The authors have not released their code.



**Strength And Weaknesses:**

Strengths
- The paper successfully demonstrates that the idea of decoupling exploration and skill discovery can be scaled to pixel-based environments.
- The proposed code resampling technique for VQ-VAE seems sensible and effective for preventing code collapse.
- The paper is well-written and easy to follow.
- The authors include some failed experiments with helpful insights in the Appendix.

Weaknesses
- One of the main weaknesses of this paper is its novelty. Their proposed method is essentially running EDL (Campos et al., 2020) on top of a discrete latent world model (Hafner et al., 2021). The notion of decoupling exploration and skill discovery is also previously suggested by Campos et al. (with the same clustering method of VQ-VAE). While their code resampling technique for preventing code collapse in VQ-VAE seems to some degree novel and sensible, it does not appear to be the primary contribution of the paper, and the degree to which it affects the final performance on downstream tasks is not discussed.
- The comparisons in the main results (Fig. 3, Table 2, and (possibly) Fig. 4 as well) seem to be not fair. They directly take the reported results from the CIC paper for baselines. However, the other skill discovery methods in the CIC paper are fine-tuned without a hierarchical controller, unlike Choreographer. How well does Choreographer perform with the same fine-tuning configuration without a controller (or the other way around)?
- In the pixel-based URLB result (Fig. 4), some of the baselines (e.g., CIC) are missing compared to Fig. 3. Also, if the numerical results are directly taken from Rajeswar et al., these comparisons could also be unfair due to the absence of hierarchical controllers for the other methods.
- It would be better if the authors can provide additional comparisons between Choreographer and LEXA (Mendonca et al., 2021), a similar skill discovery method that also uses a world model with separated exploration.
- The code is not publicly available.

Minor questions and suggestions
- How many training epochs (and environment steps) are used for the pre-training/fine-tuning of Choreographer, respectively?
- Appendix: $3e^{-4} \to 3 \cdot 10^{-4}$, etc.

**Summary Of The Paper:**

The paper proposes an unsupervised skill discovery algorithm based on state clustering in a latent space. Specifically, assuming that exploratory data are given (either by an offline dataset or by a separate exploratory policy), their proposed method learns a latent world model and clusters the latent states into $N$ discrete components using VQ-VAE. It then learns skill policies to reach the clustered states by minimizing the $\ell_2$ distance in the latent space. The paper demonstrates that their method combined with a hierarchical controller outperforms existing skill discovery methods on DM Control and MetaWorld environments.

**Summary Of The Review:**

The paper demonstrates that Choreographer can learn skills from data in pixel domains and they can be effectively adapted to downstream tasks. However, given its limited novelty and insufficient empirical comparisons with other methods, I cannot recommend acceptance at this point.

**[Post-rebuttal update]** I thank the authors for the detailed response. As my initial concerns regarding the experimental results are mostly addressed, I raised my score to 6.

---

> ### Author Response · Authors · 2022-11-19
> **Reply to Reviewer 2p4W (pt 1)**
>
> We thank the reviewer for the feedback and for highlighting that our paper presents effective ideas and is well-written.
>
> The reviewer presented some concerns related to:
> 1) analysis of the impact of code resampling on the downstream task performance
> 2) analysis of the performance without the meta-controller, both in URLB from states and from pixels
> 3) similarity of the work to a combination of Dreamer (Hafner et al, 2020) and EDL (Campos et al, 2020)
> 4) more in-depth comparison with LEXA (Mendonca et al, 2021)
> 5) availability of the code
> 6) minor changes
>
> **(1 and 2) Code resampling and meta-controller ablations**
>
> In the offline URLB from states, we investigate the usefulness of the meta-controller and compare again with CIC, the second best method on the benchmark: https://imgur.com/a/KxzkIdx.
> When removing the meta-controller, we adapt instead only one of the skills, which is selected to be the one with the highest expected rewards, considering the states and rewards obtained in the initial 4k steps of the fine-tuning stage. We see the performance remains unchanged in the Walker and Quadruped tasks, where the agent mostly requires a good initialization to master the downstream tasks quickly. In the Jaco tasks, instead, we see that having a meta-controller that considers all the skills is still beneficial. We believe this is because the task is sparser and maintaining more skills available helps exploring the environment better during fine-tuning.
>
> In the parallel exploration URLB from pixels, we investigate the usefulness of the meta-controller and of code resampling: https://imgur.com/a/GTsDLkD. We find that Choreographer always improves performance with respect to its ablations. In particular, both using the meta-controller and making sure that all the skills are active and properly learned, seem to be useful in the Walker tasks, especially in the first 500k steps, and in the Jaco tasks, at all fine-tuning times. We also note that the performance without the meta-controller remains superior to the other baselines in the URLB from pixels (from Rajeswar et al, 2022).
>
> The above ablations have been included in the updated manuscript's supplementary material (purple text).
>
> **(3) EDL + Dreamer "ablation"**
>
> In order to address the reviewer's concerns about the similarity with EDL, we have added an in-depth comparison in the Additional Related Work section of the supplementary material, in the updated manuscript (in purple text).
>
> In order to show the relevance of the differences between Choreographer and a combination of EDL + Dreamer, we have run additional experiments on URLB. In these experiments, EDL + Dreamer can be considered an ablation of Choreographer where we: (i) use EDL’s objective instead of Choreographer's  to learn the skills in imagination, (ii) remove code resampling, (iii) remove the meta-controller.
> Noting that EDL uses the mutual information between skills and environment states, compared to Choreographer which uses the mutual information between skills and latent model states, we have also experimented with a Latent EDL configuration, that uses EDL's objective but with latent states rather than environment states. Compared to Choreographer, Latent EDL still uses no entropy term in the skill reward, no meta-controller and no code resampling.
>
> **Results on URLB from states:** https://imgur.com/a/Rna7RFE
>
> In URLB from states, the performance of EDL is inferior to Latent EDL, showing that using latent codes is beneficial and Latent EDL is inferior to Choreographer, showing the usefulness of code resampling and the meta-controller.
>
> **Results on URLB from pixels (only 100k):** https://imgur.com/a/oWBaC5S
>
> In URLB from pixels, we have to note that the training time of EDL, which uses a convolutional autoencoder for the image inputs, requires ~10x more time than Latent EDL and Choreographer, as it requires reconstructing images in imagination to compare them with the decoded codes. Performance-wise, Choreographer is superior to Latent EDL, which still performs better than EDL. Given the long training time required for EDL, we could not complete training during the rebuttal period and so we will include complete results in the final version of the paper.
>
> The more detailed comparison and the empirical studies should further support the relevance of the novelties we introduced with Choreographer in order to obtain outstanding performance.

---

> > ### Author Response · Authors · 2022-11-19
> > **Reply to Reviewer 2p4W (pt 2)**
> >
> >
> > **(4) LEXA comparison**
> >
> > We included an in-depth comparison with the LEXA agent in the Additional Related Work section of the updated manuscript (purple text). The comparison can be summarized as:
> >
> >
> > |  | Choreographer | LEXA |
> > | -------- | -------- | -------- |
> > | Designed for     | fast adaptation     | zero-shot goal achievement     |
> > | Skills/goals | discrete skills sampled internally by the agent | image goals sampled from an external replay buffer |
> > |Objective| mutual information between skill and latent model states | cosine or temporal distance in image embedding space |
> > | Meta-controller| yes | no |
> >
> > **(5) Code availability**
> >
> > We release the code for Choreographer with our revision.
> >
> >
> > **(6) Minors**
> >
> > We detailed the number of training steps in Appendix B.2, which is 1 update every 10 actions, for a total of 200k updates for 2M frames/steps.
> > We also adjusted the notation from e.g. $1e^6$ to $1\cdot10^6$.
> >
> > We hope this revision improves the quality of our work and presentation.

---

### Official Review · Reviewer_LVu4 · 2022-10-24

**Confidence:** 4
**Correctness:** 3
**Technical Novelty And Significance:** 3
**Empirical Novelty And Significance:** 3
**Recommendation:** 8

**Clarity, Quality, Novelty And Reproducibility:**

Overall, this work is a clear and effective usage of mutual information skills with a world model, with interesting experiments and success when using a world model. This intuitive combination does not appear to have been experimented with before, and the necessary features to make it work are well explained. While the ablatives leave some gaps in terms of how the performance is gained, the results are promising.

Can the skills be equally as effectively learned without a clustering step and a discretized space?

While the results suggest that Choreographer can exploit data, from the introduction it seemed as though the algorithm was supposed to be exploring for its own data, which is common among mutual-information unsupervised skill learning algorithms. Choreographer can use exploration policies but does not actually introduce a novel exploration strategy on its own. However, because a world model can be sensitive to the data used to train it, it is plausible that Choreographer is actually more sensitive to the exploration strategy.


**Strength And Weaknesses:**

Latent codes based on a world model could be especially sensitive to domains where the world model struggles to generate a consistent state. Is there a way of adding information back for skill learning if it is not captured in the world model?

The experiments are somewhat weak because the experimental domains have relatively achievable modeling, and the baselines do not really make use of model-based learning. Since Choreographer makes use of imagination in the model, it will have an advantage in sample efficiency without utilizing the latent space innovation. Furthermore, the goal-reaching tasks probably should have utilized some form of hindsight, which would have much improved the baseline.

One important check would be to ensure that the latent space learned by the choreographer is giving the benefit since it appears to be the key innovation of this work. As it is, it seems like the latent code could be learned from the base state instead of the latent space of the model to get the same results, as long as the additional code resampling and learning in imagined states were implemented.

**Summary Of The Paper:**

Use unsupervised skill learning with mutual information latent space skills, but derive the skill latent space from the latent space of a world model. The world model is represented with a recurrent neural network (GRU), where the internal state of the world model is the recurrent state of the GRU, and trained to match the input data. The skills are encoded and then quantized into a fixed codebook, and unused quantizations (based on the frequency that a code is assigned) are reassigned according to an embedding from the training batch based on the distance from the closest code. The skills are trained with a reward function that combines an entropy term with a likelihood code, and the meta critic and actor chooses skill-latents as actions to optimize task reward.

**Summary Of The Review:**

I recommend acceptance as long as this method of using latent codes from aw world model has not been done in prior work, which appears to be the case.

---

> ### Author Response · Authors · 2022-11-19
> **Reply to Reviewer LVu4 (pt 1)**
>
>
> We thank the reviewer for the feedback and for highlighting the novelty and effectiveness of our work.
>
>
> > One important check would be to ensure that the latent space learned by the choreographer is giving the benefit since it appears to be the key innovation of this work. As it is, it seems like the latent code could be learned from the base state instead of the latent space of the model to get the same results, as long as the additional code resampling and learning in imagined states were implemented.
>
> Casting the skill learning problem as maximizing the mutual information between latent model states and skills, rather than between environment states and skills, is particularly useful in some settings, e.g. with pixel-based inputs. To provide empirical proof of this, we have run additional experiments where we compare Choreographer to model-based versions of:
> *  *EDL (Campos et al, 2020):* a skill learning method that extracts skills using a VQ-VAE that autoencodes environment states into codes and viceversa. The skill learning objective is the distance between decoded codes and environment states.
> *  *Latent EDL:* a modification of EDL that is closer to Choreographer, as the VQ-VAE autoencodes latent model states into codes and viceversa. The skill policies objective is now the same as the $r_\text{code}$ term in Choreographer.
>
> The experiment can be seen as an ablation analysis of Choreographer as we implement both methods using the same model-based agent of Choreographer, able to learn skills in imagination. The only other differences between Choreographer and Latent EDL lie in the *code resampling* technique, which is disabled for both the above methods, and in the *meta-controller*, which is not used by Latent EDL and EDL, as they are limited to fine-tune only one skill during adaptation time. The skill adapted is the one with the highest expected rewards, considering the states and rewards obtained in the initial 4k frames/steps.
>
> **Results on URLB from states:** https://imgur.com/a/Rna7RFE
>
> In URLB from states, the performance of EDL is inferior to Latent EDL, showing that using latent codes is beneficial and Latent EDL is inferior to Choreographer, showing the usefulness of code resampling and the meta-controller.
>
> **Results on URLB from pixels (only 100k):** https://imgur.com/a/oWBaC5S
>
> In URLB from pixels, we have to note that the training time of EDL, which uses a convolutational autoencoder to work with image inputs, requires ~10x more time than Latent EDL and Choreographer, as it requires reconstructing images in imagination to compare them with the decoded codes. Performance-wise, Choreographer is superior to Latent EDL, which still performs better than EDL. Given the long training time required for EDL, we could not complete training during the rebuttal period and so we will include complete results in the final version of the paper.
>
> We conclude that latent codes are useful both to reduce training time and to improve performance, especially when learning from high-dimensional pixel inputs. However, the outstanding performance of Choreographer also originates from other features, such as code resampling and the meta-controller.

---

> > ### Author Response · Authors · 2022-11-19
> > **Reply to Reviewer LVu4 (pt 2)**
> >
> >
> > > Latent codes based on a world model could be especially sensitive to domains where the world model struggles to generate a consistent state. Is there a way of adding information back for skill learning if it is not captured in the world model?
> >
> > As mentioned in the Limitations paragraph of the paper, we acknowledge that Choreographer's skills depend on the world model learned. We leave the idea of incorporating domain-specific information into the model or the skill representation for future work.
> >
> > > The experiments are somewhat weak because the experimental domains have relatively achievable modeling, and the baselines do not really make use of model-based learning. Since Choreographer makes use of imagination in the model, it will have an advantage in sample efficiency without utilizing the latent space innovation.
> >
> > As we state in the Baselines paragraph (Section 4), for the pixel-based URLB, we compare to the model-based baselines from (Rajeswar et al, 2022). These combine unsupervised RL algorithms with the Dreamer model-based agent, which allows learning behavior in imagination. In this setting, Choreographer achieves the overall best performance over time, showing that the skill learning and adaptation strategies of Choreographer are beneficial, compared to the other model-based agents.
> >
> >
> > > Furthermore, the goal-reaching tasks probably should have utilized some form of hindsight, which would have much improved the baseline.
> >
> > In the sparse goal-reaching tasks (Jaco and MetaWorld, Section 4.2), we do not test for the ability of the agent to reach a *given* goal in the environment, where a form of hindsight would have been beneficial. Instead, we test whether the pre-trained behavior of the agent crosses the goal position in the environment, *without having any information* about the goal and without any adaptation. In order to make this clear, we updated the manuscript stating the fact that the goal is unknown to the agent, both in the main text and in the supplementary material (purple text).
> >
> > > Can the skills be equally as effectively learned without a clustering step and a discretized space?
> >
> > We experimented with continuous skill spaces and larger discrete skill spaces as well, finding them more difficult to adapt during the fine-tuning stage, under the same computational budget. We added some extra details about this in the updated supplementary material (see the "Other things we tried" section).
> >
> > > Choreographer can use exploration policies but does not actually introduce a novel exploration strategy on its own. However, because a world model can be sensitive to the data used to train it, it is plausible that Choreographer is actually more sensitive to the exploration strategy.
> >
> > As we discuss in the Limitations paragraph of the paper, Choreographer is a data-driven approach and so the quality and the diversity of the data employed during training can affect the skill learning process. In the Additional Results section of the supplementary material, we presented a comparison of Choreographer trained with data from different ExORL datasets (Figure 9). We show that when the data has sufficient "quality", e.g. covers important states for the downstream tasks, the skills allow for fast adaptation. However, if the quality of the data is insufficient, as for ICM data in the Quadruped domain or random actions data, the skills adaptation performance is also affected.
> >
> > We hope this revision improves our paper.

---

### Official Review · Reviewer_eUdn · 2022-10-25

**Confidence:** 3
**Correctness:** 4
**Technical Novelty And Significance:** 3
**Empirical Novelty And Significance:** 3
**Recommendation:** 8

**Clarity, Quality, Novelty And Reproducibility:**

The paper is clearly written, and provides a clear discussion of how the results were obtained.

**Strength And Weaknesses:**

Choreographer is an interesting extension and combination of skill-learning algorithms and world modeling imagination approaches.  Although the combination of two existing approaches is not completely surprising, it is a novel combination as far as I am aware.  And, the method performs well against some strong baselines.  Additionally, there were some non-obvious contributions regarding code resampling in the VQ-VAE component of the world model.  Overall the paper is good.  My main concern with the paper is comparison to other methods for offline skill discovery.  For example, [Ajay et al. OPAL: Offline Primitive Discovery for Accelerating Offline Reinforcement Learning, ICLR 2021] uses a VAE to encode offline trajectories and learns skills from that data as well.  Although there are methodological differences, it would be nice to see a comparison to similar offline skill learning approaches.

**Summary Of The Paper:**

The paper presents Choreographer, a method for learning skills in the imagination of a world model.  The method uses mutual-information maximization between skill codes and latent states in a Dreamer-V2 world model.  Since the skills are learned in imagination, the data collected from the environment needs only be used for training the world model, and so Choreographer can be applied to offline RL datasets.  Once the world model and skills have been learned, Choreographer next learns a controller that uses the skills as an action space to adapt to downstream tasks.  Empirical results show that Choreographer compares favorably to other unsupervised RL methods on the URL Benchmark.

**Summary Of The Review:**

This is a strong paper that presents an interesting extension of skill learning and model-based approaches.  Although the contribution is mostly a combination of other existing work, the combination is interesting and non-trivial, and the results are strong.

---

> ### Author Response · Authors · 2022-11-19
> **Reply to Reviewer eUdn**
>
> We thank the reviewer for the feedback and for highlighting the novelty of our ideas and the relevance of our contributions.
>
> > My main concern with the paper is comparison to other methods for offline skill discovery.
>
> Offline skill discovery methods, in the literature, have focused on learning from datasets of demonstrations, play data or highly rewarded trajectories. In all these cases, the actions of the agent already show some "structure", which the agent can compress into a latent skill representation, to decode action programs that are meaningful for future tasks. Instead, Choreographer is designed for learning from completely unsupervised and unstructured data, as the experience is collected with unsupervised exploration strategies. Nonetheless, we agree that discussing these methods is important and we added a section to do so in the updated manuscript (Section F of the supplementary material, in purple text).
>
> In order to compare performance with a method from this class of approaches, we have also tested a [SPIRL agent](https://arxiv.org/abs/2010.11944) (Pertsch et al, 2020) on the offline URLB from states setting. SPIRL shares several similarities with the work mentioned by the reviewer, [OPAL](https://arxiv.org/abs/2010.13611) (Ajay et al, 2021), and has an official open-source implementation: https://github.com/clvrai/spirl.
>
> We compare a SAC agent, fine-tuned for 100k steps without any pre-training using the SPIRL codebase, and three SPIRL agents, trained with different skill length (H) parameters. Our current results, which can be found here: https://imgur.com/a/M1yKc1I, show that SPIRL seems unable to learn meaningful skills from unsupervised data. This is suggested by two facts: (i) the SAC agent performs better than all SPIRL agents, (ii) the SPIRL agent with a shorter skill length (H=2) performs better than the others. This suggests that the skills extracted do not provide coherent sequences of actions that can be leveraged by the agent during fine-tuning for solving unseen tasks.
>
> At the current status, the results of SPIRL lag behind all the other baselines in URLB from states. We signal that we have tested several hyperparameters during the rebuttal period and we will keep experimenting in the coming weeks, to include the method as a high-quality baseline in our URLB from states benchmark for the final version of our paper.
>
> We hope this revision improves the quality of our work.

---

### Official Review · Reviewer_g6c7 · 2022-10-25

**Confidence:** 4
**Correctness:** 4
**Technical Novelty And Significance:** 2
**Empirical Novelty And Significance:** 3
**Recommendation:** 6

**Clarity, Quality, Novelty And Reproducibility:**

- The paper is clear, well-structured, and easy to follow.
- The overall quality of the work is good.
- The algorithmic novelty of this work is somewhat limited, but the empirical findings are still worthwhile.

**Strength And Weaknesses:**

Strengths

- The proposed method is well-defined and quite general (e.g., the applicability to both state and visual observation environments, the VQ-VAE with the code resampling, etc.).
- The empirical evaluation and analyses, including the performance plots and tables, and qualitative results are extensive and informative.
- The high-quality presentation, including the writing and the intuitive figures (e.g., Fig.1 and Fig.2).

Weaknesses

- The work is novel to some degree, but algorithmic novelty of the work is not very strong.
- The proposed code resampling introduces an additional hyperparameter $M$, and the robustness of the method to the choice of $M$ is not examined.
- The increased burden in the size of the models (and thus the memory) and the computation in comparison with prior skill discovery methods is not discussed.

**Summary Of The Paper:**

The authors propose a model-based unsupervised skill discovery method that discovers and learns skills in the latent space using a trained world model rather than in the environment using raw observations. During the training, they keep a skill codebook equipped with a VQ-VAE for the skill encoding and propose the code resampling technique to prevent the index from collapsing. They learn skills using the mutual information maximization objective where the entropy is estimated using the particle-based estimator. They show the empirical results that the proposed method performs competitively with existing skill discovery methods.

**Summary Of The Review:**

The proposed method is somewhat limited in terms of the algorithmic novelty, but it is generally applicable, which is also supported by their empirical results. The overall quality of the work, including the experimental results and the presentation, is good. There are some weaknesses, including the lack of analysis of the method's robustness to the new hyperparameter and discussion regarding the training cost compared to prior skill discovery methods.

---

> ### Author Response · Authors · 2022-11-19
> **Reply to Reviewer g6c7**
>
> We thank the reviewer for the feedback and for highlighting that our method is quite general, has strong performance, and is well-presented.
>
> > Algorithmic novelty of the work is not very strong
>
> In our work, we proposed the novel idea of combining world models with representation learning for learning and adapting skills in imagination. In terms of algorithmic novelty, we present: (i) a new intrinsic reward formulation for skill learning, derived from the mutual information between latent model states and skills (rather than environment states and skills), (ii) *code resampling*, a technique to address the issue of index collapse in VQ-VAEs.
>
> > The proposed code resampling introduces an additional hyperparameter, and the robustness of the method to the choice of the hyperparameter is not examined.
>
> We ran an ablation of different values of $M$ ("resample every $M$ updates) with a codebook dimensionality of 512 so that the impact of code resampling is noticeable (as shown in Figure 7). We present the results at https://imgur.com/a/3xccPvj and in the updated manuscript (new text in purple). Overall, the technique is robust to a large range of hyperparameters, as the representation converges to similar values of reconstruction loss and % of unused codes over time. However, we suggest setting $M$ to lower values, e.g. 10 or 100, in contexts where earlier convergence can be useful.
>
> > The increased burden in the size of the models (and thus the memory) and the computation in comparison with prior skill discovery methods is not discussed.
>
> In our work, we mostly focus on data-efficiency, as Choreographer is designed to have strong performance in fine-tuning low-data regimes. However, as this can be useful for future readers, we thank the reviewer for the suggestion and we signal that we added a discussion on the additional parameter and computational efficiency in the supplementary material of the updated manuscript (Section B.4 "Considerations about efficiency").
>
> We hope this revision improves the quality of our work.

---

### Author Response · Authors · 2022-11-28
**General comment**

We would like to thank all reviewers for the time they spent reviewing our work and for providing constructive feedback.

We are glad that all reviewers found our approach significant and the work of high quality. We addressed the concerns raised by the reviewers in the individual responses and we revised our manuscript according to the suggestions provided. Here we summarize the main changes:

* (**all**) we released the code in the Supplementary Material
* (**g6c7**) we included an ablation about our code resampling technique in the Appendix
* (**g6c7**) we added a discussion on data, computation, and parameter efficiency in Section B.4 ("Considerations about efficiency") in the Appendix
* (**eUdn**) we added a related work section about offline skill learning in Section F ("Additional Related Work") in Appendix
* (**eUdn**) to answer the reviewer's concerns, we ran experiments using the SPIRL offline skill learning approach (https://github.com/clvrai/spirl) in the same setting of Choreographer (offline state-based URLB). Our initial results show the approach seems to struggle when learning from completely unsupervised data. We will include an extensive comparison in the final version of the paper.
* (**LVu4, 2p4W**) we posted additional "ablations" of our method, showing the importance of some of the components we introduce, namely: (i) the mutual information with latent states objective, (ii) the meta-controller, (iii) code resampling.
* (**2p4W**) we included an in-depth comparison with EDL (Campos et al, 2020) and LEXA (Mendonca et al, 2021) in Section F ("Additional Related Work") in Appendix

We hope our revision addresses the raised concerns and we are happy to incorporate additional feedback or respond to further questions.

---

### Decision · Program_Chairs · 2023-01-20

**Decision:**

Accept: notable-top-25%

**Justification For Why Not Higher Score:**

Algorithmic novelty is limited as the approach presents a combination of known algorithms and ideas.

**Justification For Why Not Lower Score:**

Interesting ideas, challenging problem setup and extensive evaluation.

**Metareview: Summary, Strengths And Weaknesses:**

The paper presents a new algorithm to discover skills based on latent imagination. The proposed approach uses the dreamer v2 model and use a  a skill codebook equipped with a VQ-VAE for the skill encoding. The authors propose the code resampling technique to prevent the index from collapsing. The skills are learned using a mutual information objective which is common to unsupervised skill discovery.

The paper is well motivated, it presents an interesting idea and addresses  a challenging problem, i.e. unsupervised skill discovery in image based RL domains. Experiments are also extensive and convincing. In terms of limitations, the reviewers were concerned about the limited novelty as the algorithm is basically a combination of existing approaches and ideas. Yet, all reviewers found the contribution and idea to be significant and evaluated the paper positively. I agree with their judgement.

**Note From Pc:**

if the above contains the word "oral" or "spotlight" please see: "oral" presentation means -> notable-top-5% and "spotlight" means -> notable-top-25%. As stated in our emails, we are disassociating presentation type from AC recommendations

**Summary Of Ac-Reviewer Meeting:**

N/A